# Chemogenomic profiling to understand the antifungal action of a bioactive aurone compound

Fatmah M. Alqahtani[1], Brock A. Arivett[1], Zachary E. Taylor[2], Scott T. Handy[2], Anthony L. Farone[1], Mary B. Farone[1]*

1 Department of Biology, Middle Tennessee State University, Murfreesboro, Tennessee, United States of America, 2 Department of Chemistry, Middle Tennessee State University, Murfreesboro, Tennessee, United States of America

* mary.farone@mtsu.edu

**Data Availability Statement:** The sequence data are publicly available in the NCBI SRA under BioProject accession number PRJNA491750. Perl scripts are available from GitHub at

## Abstract

Every year, more than 250,000 invasive candidiasis infections are reported with 50,000 deaths worldwide. The limited number of antifungal agents necessitates the need for alternative antifungals with potential novel targets. The 2-benzylidenebenzofuran-3-(2H)-ones have become an attractive scaffold for antifungal drug design. This study aimed to determine the antifungal activity of a synthetic aurone compound and characterize its mode of action. Using the broth microdilution method, aurone SH1009 exhibited inhibition against *C. albicans*, including resistant isolates, as well as *C. glabrata*, and *C. tropicalis* with $IC_{50}$ values of 4–29 µM. Cytotoxicity assays using human THP-1, HepG2, and A549 human cell lines showed selective toxicity toward fungal cells. The mode of action for SH1009 was characterized using chemical-genetic interaction via haploinsufficiency (HIP) and homozygous (HOP) profiling of a uniquely barcoded *Saccharomyces cerevisiae* mutant collection. Approximately 5300 mutants were competitively treated with SH1009 followed by DNA extraction, amplification of unique barcodes, and quantification of each mutant using multiplexed next-generation sequencing. Barcode post-sequencing analysis revealed 238 sensitive and resistant mutants that significantly (FDR *P* values ≤ 0.05) responded to aurone SH1009. The enrichment analysis of KEGG pathways and gene ontology demonstrated the cell cycle pathway as the most significantly enriched pathway along with DNA replication, cell division, actin cytoskeleton organization, and endocytosis. Phenotypic studies of these significantly enriched responses were validated in *C. albicans*. Flow cytometric analysis of SH1009-treated *C. albicans* revealed a significant accumulation of cells in G1 phase, indicating cell cycle arrest. Fluorescence microscopy detected abnormally interrupted actin dynamics, resulting in enlarged, unbudded cells. RT-qPCR confirmed the effects of SH1009 in differentially expressed cell cycle, actin polymerization, and signal transduction genes. These findings indicate the target of SH1009 as a cell cycle-dependent organization of the actin cytoskeleton, suggesting a novel mode of action of the aurone compound as an antifungal inhibitor.

https://github.com/fma3b/Barcode_Seq_Analysis.
All other relevant data are within the paper and its
Supporting Information files.

**Funding:** This research was supported in part by
the Tennessee Center for Botanical Medicine
Research and the Molecular Biosciences Program
at Middle Tennessee State University. The funders
had no role in the study design, data collection and
analysis, the decision to publish, or manuscript
preparation.

**Competing interests:** The authors have declared
that no competing interests exist.

## Introduction

Life-threatening fungal infections have been increasing due to the difficulties with diagnosis
and treatment that accelerate mortality rates associated with fungal infections, which now
exceed deaths caused by malaria [1]. *Candida albicans* is the most frequently isolated opportunistic fungal pathogen and is implicated in superficial mucosal infections, or candidiasis of the
oral cavity and genitalia of humans, particularly in immunocompromised patients [2]. In
healthy individuals, *Candida* spp. are a commensals of the mucosal surfaces of genitalia, oral
cavity, and gastrointestinal tract. However, with the introduction of antibacterial antibiotics as
medical therapy in the 1940s, a gradual increase in the number of invasive candidiasis cases
has been reported due to antibiotic-associated loss of the bacterial biota and subsequent colonization of *Candida* spp. on epithelial surfaces, a requirement for pathogenesis [3]. Several risk
factors contribute to the pathogenesis of invasive candidiasis, including organ transplantation,
prolonged hospitalization in an intensive care unit, catheterization, and intensive utilization of
antibiotics and immunosuppressive agents. These factors could lead *Candida* spp. to colonize
mucosal surfaces, resulting in superficial infections. The fungus can also advance to candidemia, or invasion of the bloodstream, and from there disseminate to different organs. Certain
virulence factors are attributed to the pathogenicity of *Candida* spp., including adherence to
epithelial surfaces, dimorphic growth, biofilm formation, and production of tissue-damaging
enzymes [4, 5].

For treating candidiasis, there are five groups of antifungal agents as defined by their mode
of action and for which mechanisms of resistance have been described. Group I: polyenes
(amphotericin B) bind to ergosterol in the cell membrane and form pores in it, while Group
II: echinocandins (caspofungin) inhibit β(1,3)-glucan synthase in the cell wall. Group III:
azoles (fluconazole) inhibit lanosterol 14 α-demethylase in the ergosterol biosynthesis pathway. Group IV: synthetic pyrimidines (5-fluorocytosine) inhibit DNA synthesis and disturb
protein synthesis, and Group V: allylamines (terbinafine) inhibit squalene epoxidase in the
ergosterol biosynthesis pathway [6]. Resistance mechanisms have been described to these antifungals as cellular determinants that lead to drug extrusion by active efflux, altered drug targets, or drug target overexpression. However, novel drug resistance mechanisms have been
recently reported as robust responses that enhance antifungal tolerance by pathways such as
regulation of the oxidative or thermal stress responses [7].

Even with treatment by commercially-available antifungal agents, the mortality rate from
disseminated candidiasis has surged to ~40–60%, representing a 20-fold increase compared to
only two decades ago [3]. Every year, more than 250,000 invasive candidiasis infections are
reported with 50,000 deaths worldwide [4]. Furthermore, in the USA alone, the cost of combating candidiasis was estimated to be $2–4 billion annually in the year 2000 [8]. Candidiasis
has recently been reported as the third-to-fourth most frequent healthcare-acquired infection
globally [9]. Although the majority of candidiasis cases in humans are attributed to *C. albicans*,
other *Candida* species have not only emerged as causative agents of candidiasis but have also
developed resistance to antifungal drugs. These species most often include *C. glabrata*, *C. tropicalis*, *C. parapsilosis*, and *C. krusei* [3]. The expanding immunosuppressed population, the limited number of fundamental antifungal agents along with their resistances and toxicity issues,
and the emergence of non-albicans pathogenic strains all necessitate the need to seek alternative antifungal agents with potential novel targets.

To achieve this goal of seeking alternative antifungals, the exploitation of natural products,
particularly those derived from plants, appears to be a promising source for antifungal compound development [10]. Because plants have their own fungal pathogens, these interactions
between plants and fungi have resulted in the origination of diverse chemical entities within

**Fig 1. Chemical structure of aurone SH1009.**

the plants intended to enhance not only their protection from fungal pathogens, but their survival and competitiveness as well [11]. One of the most promising classes of natural products are the secondary metabolites, aurones, which are ubiquitously present in plants. Aurones, or 2-benzylidenebenzofuran-3-(2H)-ones, are structural isomers of flavonoids that naturally occur as yellow-color pigments in plants [12–14]. In addition to their roles in pigmentation, aurones possess a variety of protective roles in the plant, including insect antifeedant [13], antiparasitic [14], and antifungal activities [15, 16]. Moreover, activities of aurones as anticancer [16, 17], antiparasitic [14], antileishmanial [18, 19], and antifungal agents have been reported [20]. Since the bioactive properties and therapeutic prospective of natural and synthetic aurones are promising, these bioactive compounds can be considered as an attractive scaffold for antifungal drug design and development.

We have previously reported the synthesis and anti-*Candida* activities of non-natural aurone derivatives containing different functional groups, including aurone SH1009 (Fig 1), which exhibited significant inhibition of *C. albicans* when compared to the other derivatives [21]. The reported disruption of biofilm formation by aurone SH1009 emphasizes the importance of understanding the SH1009 mode of action. In the present study, the antifungal activity of aurone SH1009 was determined against different standard and clinical *Candida* spp., including resistant isolates, applying a modified synthetic strategy based on an acid-mediated condensation between the appropriate benzofuranone and aldehyde. The mode of antifungal action of aurone SH1009 was characterized using chemogenomic approach in *Saccharomyces cerevisiae* mutant collections and validated in *C. albicans* SC5314.

## Results

### Aurone SH1009 is selectively inhibitory for *Candida* spp.

To assess the antifungal activity of aurone SH1009 against *Candida* strains (listed in S1 Table), antifungal susceptibility testing was performed using the Clinical Laboratory Standards Institute (CLSI) broth microdilution protocol [22]. SH1009 exhibited promising antifungal activity

**Table 1. The means of IC$_{50}$ (inhibitory concentration that causes 50% inhibition) and MIC$_{90}$ (minimal inhibitory concentration that causes 90% inhibition) ± the SEM of aurone SH1009 for different *Candida* spp.**

| Strains | IC$_{50}$ (μM) | MIC$_{90}$ (μM) |
|---|---|---|
| *C. albicans* ATCC 90028 | 11±0.345 | 18.75±6.25 |
| *C. albicans* ATCC 90029 | 13±1.125 | 25±0 |
| *C. albicans* M1: SC5314 | 16±2.4 | 25±9.375 |
| *C. albicans* M2: ScTAC1R34A a | 11±0.725 | 18.75±6.25 |
| *C. albicans* M3: ScMRR1R34A a | 12±0.14 | 25±0 |
| *C. albicans* M4: Gu2 | 11±0.13 | 12±0 |
| *C. albicans* M5: Gu5 a | 4±1.159 | 9.375±3.125 |
| *C. albicans* M6: F1 | 10±0.54 | 12±1.3 |
| *C. albicans* M7: F5 a | 8±0.676 | 12.5±0 |
| *C. albicans* ATCC 64124 b | 21±7.125 | 50±0 |
| *C. glabrata* ATCC 66032 | <3.125±0 | <3.125±0 |
| *C. tropicalis* ATCC 750 | 29±25.7 | 62.5±37.5 |

[a] Resistant to fluconazole.

[b] Resistant to amphotericin B, caspofungin, fluconazole, and 5-fluorocytosine except at high concentration.

against all *Candida* spp. tested (Table 1). *C. albicans* strains M2, M3, M5, and M7 are all fluconazole resistant and, as such, represent a serious risk for patients since fluconazole is the principle antifungal drug for treating candidiasis [2]. In this study, fluconazole had no inhibitory activity on these isolates even at high concentrations (MIC > 64 μg/mL), while the IC$_{50}$ of aurone SH1009 is considerably lower [11, 12, 4, and 8 μM) for *C. albicans* M2, M3, M5, and M7 strains, respectively. Similarly, another aggressive, multi-drug resistant isolate, *C. albicans* ATCC 64124, exhibited resistance to a high concentrations of amphotericin B (16 μg/mL), caspofungin (4 μg/mL), and fluconazole (64 μg/mL), although 5-fluorocytosine was inhibitory (MIC < 1 μg/mL). This strain was also sensitive to SH1009 (IC$_{50}$ = 21 μM). Although *C. albicans* is recovered from at least 50% of candidiasis cases, *C. glabrata* has recently emerged as responsible for approximately 25% of cases in the last two decades, and tends to infect elderly populations [23]. Exposure of *C. glabrata* to SH1009 resulted in inhibition with a low concentration of aurone (IC$_{50}$ < 3.125 μM), suggesting it could be a successful treatment, especially since the development of *C. glabrata* resistance to caspofungin and lower susceptibility to fluconazole have been reported [24]. *C. tropicalis*, a causative agent in 10–20% of candidiasis cases in the USA and 35–40% of cases in tropical regions, was also susceptible to SH1009 [3]. Because aurone SH1009 is not affected by the resistance mechanisms of these isolates with known resistances to antifungals, the compound presents new possibilities for further exploration as a potential antifungal agent.

Although SH1009 was inhibitory, the aurone was not fungicidal even at the highest concentration of aurone tested (200 μM), as indicated by colony growth on solid media. Therefore, a more sensitive cell viability assay, compatible with yeast cells, was required to more precisely measure viable cell number after treatment. Along with flow cytometry, the live/dead Fungalight Yeast Viability Kit was used to quantitate viability after SH1009 treatment. After 48 h of treatment with a high SH1009 concentration (200 μM), *C. albicans* SC5314 strain (Fig 2A) revealed two divided subpopulations after gating for the stained cells in the dot plot graphs. When compared to isopropanol-treated cells (Fig 2B), a significant fraction (Fig 2D) of SH1009-treated cells clustered in the upper-left quadrant, indicating the bioactivity of SH1009 in eliminating *C. albicans* SC5314 growth. In contrast, the viable cells in the SH1009-treated

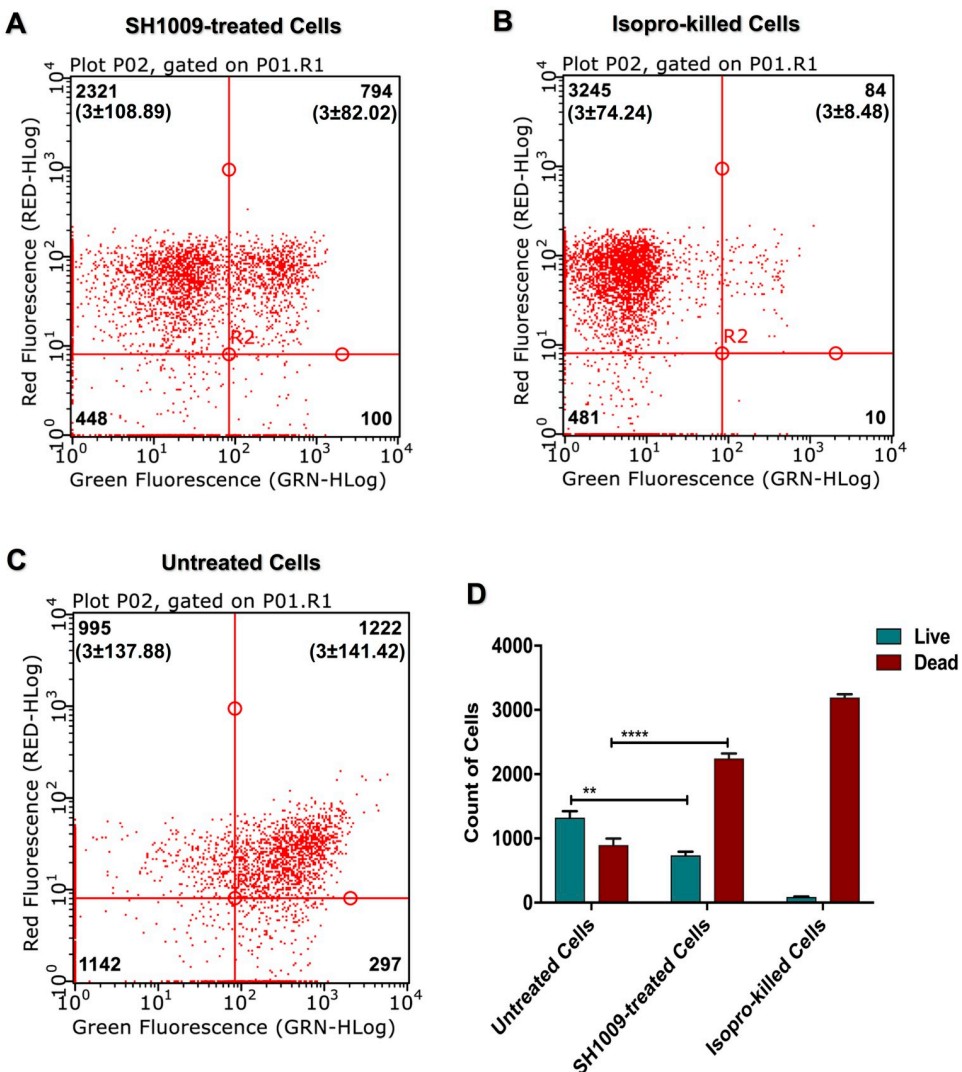

**Fig 2. Cell viability assay of aurone SH1009-treated *C. albicans*.** *C. albicans* SC5314 cell suspensions were stained with SYTO® 9 dye (green fluorescence) and Propidium Iodide (red fluorescence) and analyzed using the Millipore Guava flow cytometer and InCyte software system. For each dot plot **A)** 200 μM aurone SH1009-treated cells, **B)** isopropanol-killed cells, and **C)** untreated cells, the upper-left quadrants show the count of dead cells and the upper-right quadrants show the count of live cells. **D)** Significance was calculated using two-way ANOVA to compare the cell viability of two population groups (live and dead) between untreated and SH1009-treated cells. *P* values (**** $P \leq 0.0001$), (** $P \leq 0.01$). (n = 3±SD).

sample (Fig 2A) were reduced by approximately half compared to the untreated-cells sample (Fig 2C).

Growth of SH1009-treated *C. albicans* SC5314 was monitored at 30-min intervals over a 46 h incubation using dilutions of the aurone. A representative set of growth curves of *C. albicans* SC5314 is given in Fig 3A. The curves indicate that there is no regrowth of *C. albicans* SC5314 in the presence of SH1009 above 12.5 μM. Additionally, inhibition of *C. albicans* SC5314 by SH1009 is highly dose dependent. The potency of any bioactive-compound is usually measured by the concentration of compound that inhibits 50% ($IC_{50}$) of a fixed-pathogenic inoculum in a dose–response assay *in vitro*. However, dose–response curve slope is another clinically independent criterion that can be used as an indicator of the expected therapeutic

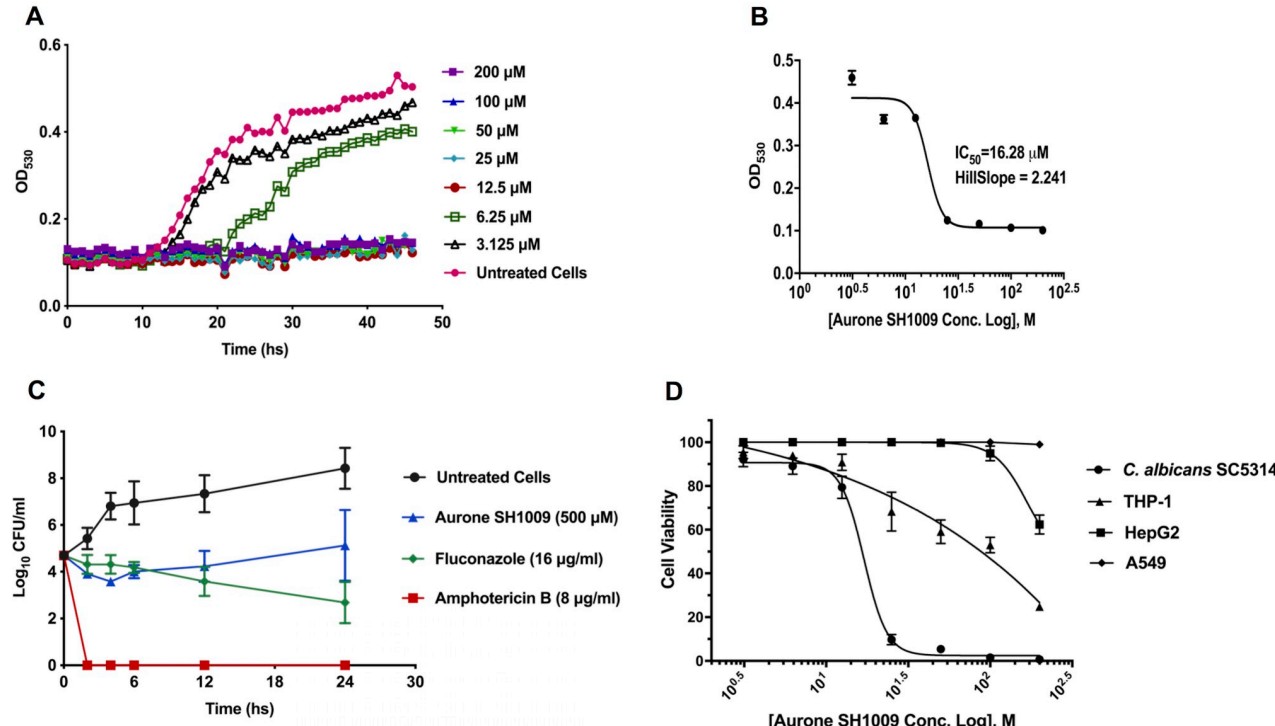

**Fig 3. Aurone SH1009 exhibited dose-dependent inhibition of growth. A)** Growth curves of *C. albicans* SC5314 strain ($2.5$–$0.5 \times 10^3$ CFU/mL) in dilutions of SH1009 ($3.125$ μM– $200$ μM) in RPMI-1640 medium using the Bioscreen C growth curve instrument to measure the $OD_{530}$ every 30 min for 46 hours at 35˚C. **B)** Graphing the nonlinear regression of $OD_{530}$ readings using GraphPad Software to calculate the $IC_{50}$ value after transforming the molar concentrations of the aurone SH1009 into the logarithmic form. **C)** Time-kill plot of *C. albicans* SC5314 cells ($0.5$–$0.25 \times 10^5$ CFU/mL) treated with concentrations five-fold higher than the $IC_{50}$ concentrations of SH1009 ($16.28$ μM), fluconazole ($0.5$ μg/mL) and amphotericin B ($0.25$ ug/mL) with incubation at 35˚C for indicated time points. Colony forming units (CFU) were determined by plating 10 μL of each treatment onto YPD agar plates at each timepoint. **D)** Cytotoxic effects of aurone SH1009 for *C. albicans* SC5314 and THP-1, HepG2, and A549 human cell lines are presented as dose-response curves by graphing the nonlinear regression of the cell viability using GraphPad Software to calculate the $CC_{50}$ values after transforming the molar concentrations of the aurone SH1009 into logarithmic form.

effectiveness [25]. Typically, a dose–response curve with a particularly steep slope (Hill coefficient >1) indicates that a small increase in concentration of a drug above the $IC_{50}$ causes extraordinarily high-level inhibition. SH1009 exhibited a steeper dose-response $IC_{50}$ curve in inhibition of *C. albicans* SC5314 as is indicated by the higher slope factor (~2.241) (Fig 3B). The steep slope of the $IC_{50}$ curve reflects the growth curve results for treated cells, in that there is significant loss of viability with aurone concentrations above the $IC_{50}$ concentration of $16.28$ μM. This predicts a possible therapeutic potency for SH1009 with a selective 50% inhibitory concentration required to inhibit *C. albicans* SC5314 cells ($0.5$–$2.5 \times 10^3$ CFU/mL).

A time-kill assay was employed to investigate the killing kinetics of aurone SH1009 against *C. albicans* SC5314 along with fluconazole and amphotericin B treatments (Fig 3C). As expected, amphotericin B affected the growth curve at 8 μg/mL after 2 h, regardless of the initial inoculum of *C. albicans* SC5314 cells ($0.5$–$0.25 \times 10^5$ CFU/mL), demonstrating fungicidal activity. Treatment with SH1009 at a concentration representing an approximate five-fold increase of the $IC_{50}$ calculated from the two-fold dilution series (500 μM) reduced the growth of the yeast similarly to fluconazole (16 μg/mL) with no significant differences until 12h of incubation, indicating fungistatic activity. Given the initial inoculum, the SH1009 treatment yielded an approximate > 3 $log_{10}$ decrease in CFU/mL after each time point of treatment compared with untreated culture, exhibiting a significant reduction ($P$ value $\leq 0.0001$) in the

colony count. This reduction in the CFU/mL resulted in no appreciably significant increase in growth rate after SH1009 treatment over 24h, whereas the untreated culture recorded $> 4.4$ $\log_{10}$ significant increase in the CFU/mL between 0 and 24h. These results indicate a fungistatic activity which might be less desirable for immunosuppressed patients to remedy opportunistic infections. Consequently, we measured the cytotoxicity of SH1009 for three different human cell lines.

Toxicity assays to determine the cytotoxic concentration of aurone that reduced cell viability by 50% ($CC_{50}$) were carried out based on the reduction of resazurin to resorufin by metabolically active cells as a sensitive method for detecting viable cells [26]. The $CC_{50}$ values for THP-1, HepG2, A549 cells after 24h treatment with two-fold serial dilutions of SH1009 (3.125–200 μM) were 140, 168, and >200 μM, respectively (Table 2). Selectivity index (SI) values were calculated to correlate the antifungal activity of aurone SH1009 against *C. albicans* SC5314 cells ($IC_{50}$ 16.28 μM) with the $CC_{50}$ concentration for the human cell lines. The $CC_{50}$/ $IC_{50}$ ratio was an ~8.6->12-fold difference in the concentrations that resulted in 50% loss of cell viability, suggesting a selectivity of SH1009 for the pathogenic yeast cells over human cells. Treatment of *C. albicans* SC5314 cells with SH1009 at 25 μM resulted in a significant reduction ($P \leq 0.001$) in cell viability when compared to the untreated control group, while treatment of human cell lines with the same concentration resulted in no appreciably significant reduction with 70% cell viability for THP-1 cells and 100% viability for both HepG2 and A549 cells. With increasing SH1009 concentrations, the cell viability of *C. albicans* SC5314 was reduced significantly ($P \leq 0.01$), compared to the cell viability of human cells (Fig 3D). These results indicate that aurone SH1009 has a selective toxicity for C. *albicans* cells. To assist with assessing the therapeutic potential of aurone SH1009 as an antifungal, we next sought to determine the bioactivity of the compound by defining its mode of toxicity.

## Chemogenomic profiling of aurone SH1009-treated yeast cells identifies roles for genes involved in the cell cycle, cell division, and the actin cytoskeleton

Chemogenomic profiling was used to characterize the mode of action for aurone SH1009. Haploinsufficiency profiling (HIP) and homozygous profiling (HOP) allow for paralleled assessment of the sensitivity and resistance of the pooled genome-wide set of *S. cerevisiae* deletion mutants. First, two pools of *S. cerevisiae* heterozygous (HIP) and homozygous (HOP) deletion mutant collections were treated with aurone SH1009 at the concentration (~500 uM) that inhibited the growth of the wild type *S. cerevisiae*-S288C ($1.25\times 10^9$ cells /mL) by 20% for 48h. After purifying genomic DNA from the mutants, the synthetic UPTAG DNA barcodes (20 bp) were amplified using uniquely indexed primers to distinguish each sample. Because each mutant is uniquely identified with DNA barcode, multiplexed-next generation sequencing as a highly robust technique was employed to quantitate the abundance of each mutant.

Haploinsufficiency profiling was performed on ~1056 heterozygous mutants that are essential for growth and express only 50% of gene dosage because one functional copy of that

Table 2. The $CC_{50}$ (cytotoxicity concentration of aurone SH1009 that causes 50% cell viability loss) ± the SEM and the selectivity index (SI) as a ratio between the $CC_{50}$ for the mammalian cells divided by the $IC_{50}$ against *C. albicans* SC5314.

| Human cell line | $CC_{50}$(μM) | SI |
|---|---|---|
| THP-1 (ATCC, TIB-202) | 140±4.5 | 8.6 |
| HepG2 (ATCC, HB-8065) | 168±7.6 | 10.31 |
| A549 (ATCC, CCL-185) | >200±0 | >12 |

particular gene in the diploid organism has been deleted. Whereby, the identification of the direct target of a certain bioactive compound can be identified in the presence of that compound as the mutant that has a large fitness defect compared to the other mutants that do not encode the drug target. Conversely, homozygous profiling was performed on ~4244 of homozygous mutants that are non-essential for growth and express 0% of gene dosage because both copies of the particular gene are deleted in the diploid organism. With the HOP assay, it is possible to suppress drug sensitivity due to the complete loss-of-function alleles, allowing identification of pathways that confer the drug sensitivity or identification of the direct target of drugs following the principle that deletion of drug target will render the cells insensitive to the compound [27].

The post-sequencing data analysis of the aurone-treated deletion mutants revealed 3923 mutants, which included 3,133 mutants from the homozygous deletion pool (non-essential genes) and 790 mutants from the heterozygous deletion pool (essential genes), representing ~ 75% of the mutant population with usable read counts and 0.90 correlation between samples, indicating high sample quality and agreement. The chemical-genetic interaction of the positive control methyl methanesulfonate (MMS), a well-characterized antifungal agent that damages DNA [28], demonstrated a highly significant enrichment ($P$ value $\leq 0.001$) for cellular response to DNA damage stimulus, confirming a successful HIP-HOP assay procedure and accurate post-sequencing data analysis. The chemical-genetic interaction identified 238 gene deletion mutants that were significantly responsive to aurone SH1009 (FDR $P$ values $\leq 0.05$) for both HIP-HOP profiles (Fig 4A and 4B). The sensitive and resistant genes with that $P$ value (approximately $\geq 1.5$ fold change) for HIP and HOP independently were used for gene ontology (GO) enrichment analysis using ClueGO [29] and Yeast Gene Ontology Slim Term Mapper at *Saccharomyces* Genome Database (SGD) [30].

For essential genes (HIP profile), GO enrichment ($P \leq 0.0001$) for mutants that have deleted genes encoding cell cycle proteins were found (e.g. *CDC28Δ*, *CDC34Δ*, *CDC25Δ*, *CDC13Δ*, *ORC4Δ*, *MCM2Δ*, *MCM7Δ*, *CBK1Δ*, and *RFC1Δ*) (Fig 4C). Moreover, two other biological responses involved in the cell cycle showed a significant GO enrichment ($P \leq 0.01$). These included genes for meiosis-yeast (*ORC4Δ*, *MCM2Δ*, *MCM7Δ*, *APC4Δ*, *APC1Δ*, *CDC28Δ*, and *CDC5Δ*) and DNA replication (*POL3Δ*, *MCM2Δ*, *MCM7Δ*, *RFC3Δ* and *RFC1Δ*). For non-essential genes (HOP profile), among both sensitive and resistant mutants, GO enrichment ($P \leq 0.01$) detected sensitive mutants (*ARK1Δ*, *NIP100Δ*, *GIC1Δ*, *SLK19Δ*, *KIP3Δ*, and *ACF4Δ*) and resistant mutants (*BOI2Δ*, *EDEΔ*, *END3Δ*, and *GIC2Δ*) with deleted genes that are associated with actin cytoskeleton organization (Fig 3D). Also, mutants that are associated with cell division were significantly enriched from the HOP profile (*LDB19Δ*, *SLK19Δ*, *LTE1Δ*, *ZIP2Δ*, *BUB1Δ*, *ELM1Δ*, *TOF1Δ*, *TOF2Δ*, *SWI4Δ*, *CSM3Δ*, *CLB3Δ*, and *CLN2Δ*). Lastly, there was an enrichment of deletion mutants for the biological response of endocytosis, which also relates to the actin cytoskeleton, that included mutants that were among the top 10 sensitive in the HOP profile (*FEN2Δ*, *LTE1Δ*, *LDB19Δ*, and *ARK1Δ*) (S2 and S3 Tables).

For several phenotypes described above, the cell cycle pathway appeared to be the primary target pathway that was significantly enriched by SH1009 treatment for the essential genes profile and non-essential genes profile, separately. To obtain a broader view of the changes in growth patterns of heterozygous and homozygous mutants after SH1009 treatment, the significantly responsive mutants from both HIP-HOP profiles were combined and used for the enrichment analysis. For all 238 deleted protein-coding genes with (FDR $P$ values $\leq 0.05$ and fold change $\geq 1.5$) from both profiles, the clustered analysis of KEGG pathways and gene ontology terms that significantly enriched (FDR P values $\leq 0.05$) were plotted for the number of genes that are associated with each term in a histogram (Fig 5). Notably, both the profiles for the 80 essential genes (HIP-profile) and 158 non-essential genes (HOP-profile) are largely

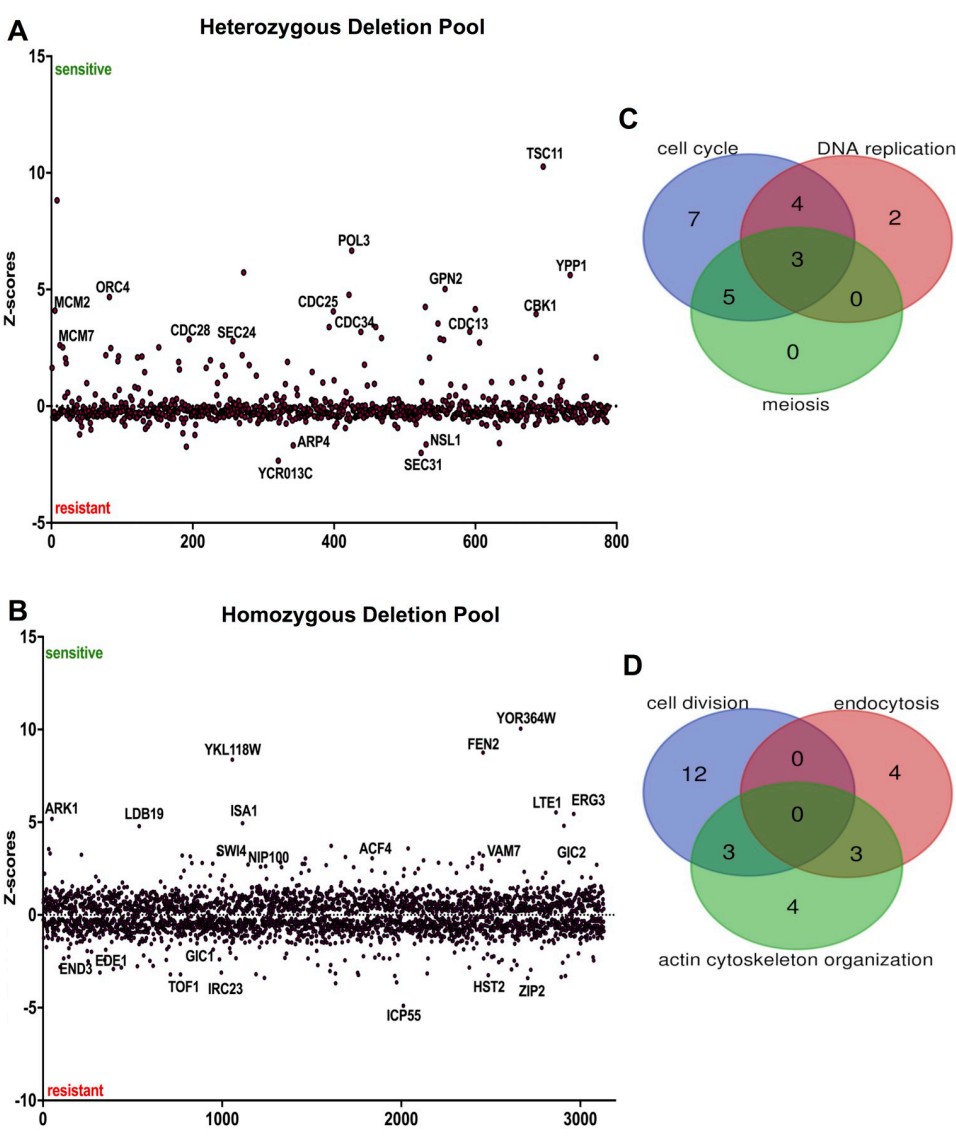

**Fig 4. Chemical-genomic analysis of aurone SH1009. A)** Z-score plot of heterozygous deletion pool (HIP profile) for essential genes, and **B)** Z-score plot of homozygous deletion pool (HOP profile) for non-essential genes where the sensitive mutants have positive scores and the resistant mutants have negative scores. **C)** and **D)** Number of mutants whose sensitivities or resistances were affected by aurone SH1009 (FDR $P \leq 0.05$, fold change $\geq 1.5$) clustered by significant biological responses and represented in the Venn diagram for HIP and HOP profiles, respectively.

clustered for most the significantly enriched category (S4 Table). These significantly responsive genes were mapped to six pathways in the KEGG pathway database, with the cell cycle pathway as the most significantly enriched pathway ($P \leq 0.0001$) along with other pathways that are completely overlapped with cell cycle pathway (meiosis and DNA replication). For gene ontology categories, within the biological process category 18 terms were enriched in differentially sensitive or resistant mutants, including cell division, cytoskeleton organization, and regulation of endocytosis. Also, the nucleotide-binding, aminoacyl-tRNA ligase activity, and DNA-binding terms were significantly enriched in the molecular function category. Nucleus, cytoskeleton, cellular bud, as well as the site of polarized growth terms were significantly enriched in the cellular component category.

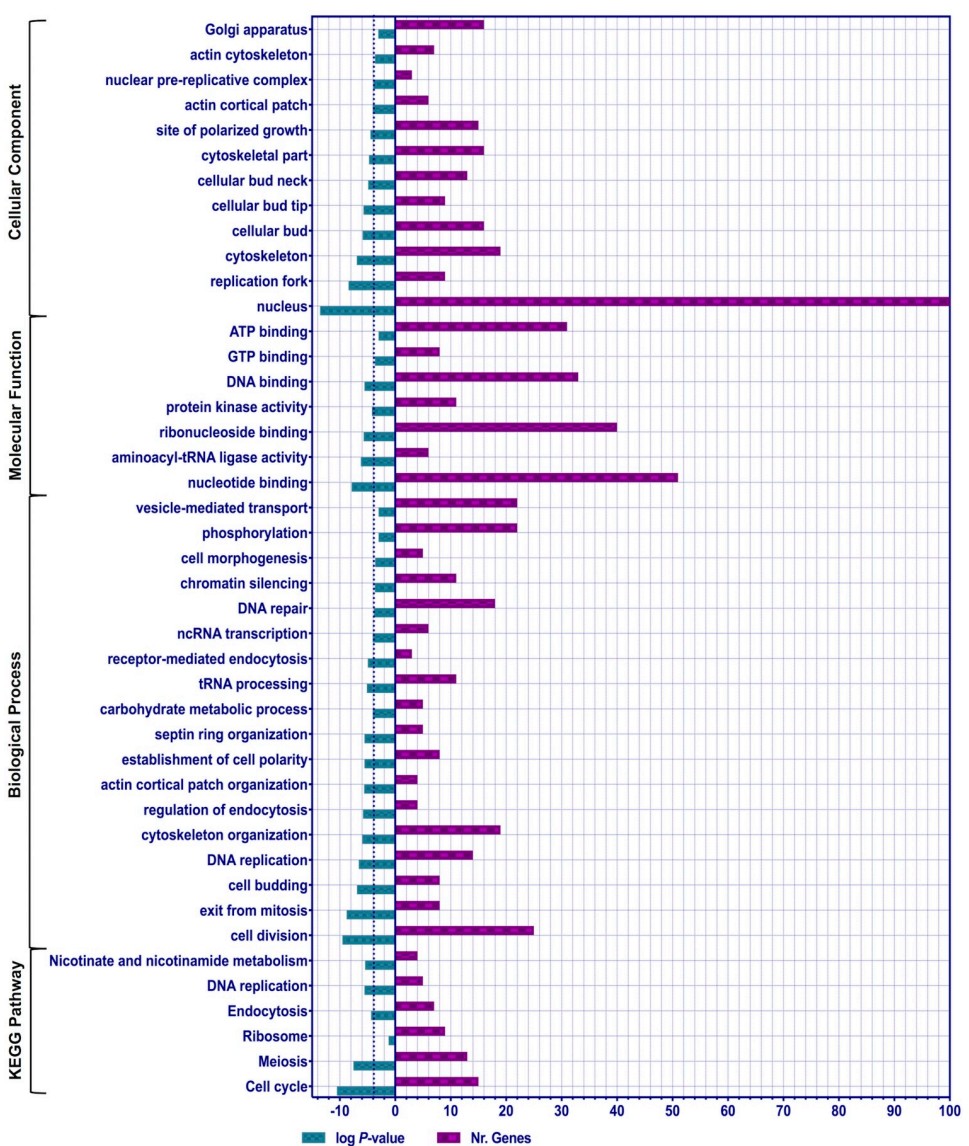

**Fig 5. Functional categories of KEGG pathway and gene ontology (biological process, molecular function, and cellular component) enrichment analysis of 238 differentially responsive mutants with FDR ≤ 0.05 and fold change ≥ 1.5 from heterozygous deletion pool (HIP) and homozygous deletion pool (HOP) profiles.** Purple bars represent the number of genes, 80 essential genes from HIP-profile and 158 non-essential genes from HOP-profile, that are clustered in each KEGG/GO term. Green bars show the significance of each category as log *P*-values that are calculated by hypergeometric calculation through ClueGo software using GO categories in the *Saccharomyces cerevisiae*-S288C as a background with an FRD ≤ 0.05 as a cutoff significant value for all the plotted categories. The threshold-dotted line depicts the highly significant KEGG/GO categories with *P* ≤ 0.01.

In order to gain insight into the chemical-genetic interaction of genes targeted by aurone SH1009 and visualize the connections between targeted biological responses and pathways identified by both the HIP and HOP profiles, ClueGO and CluePedia along with Cytoscape software were used to extract and map non-redundant biological responses for a large set of functionally clustered genes using GO terms and KEGG pathway, simultaneously [29, 31]. Fig 6A depicts the functional annotation network of clustered essential and non-essential genes. This overlapped network revealed relatively connected biological categories that started from

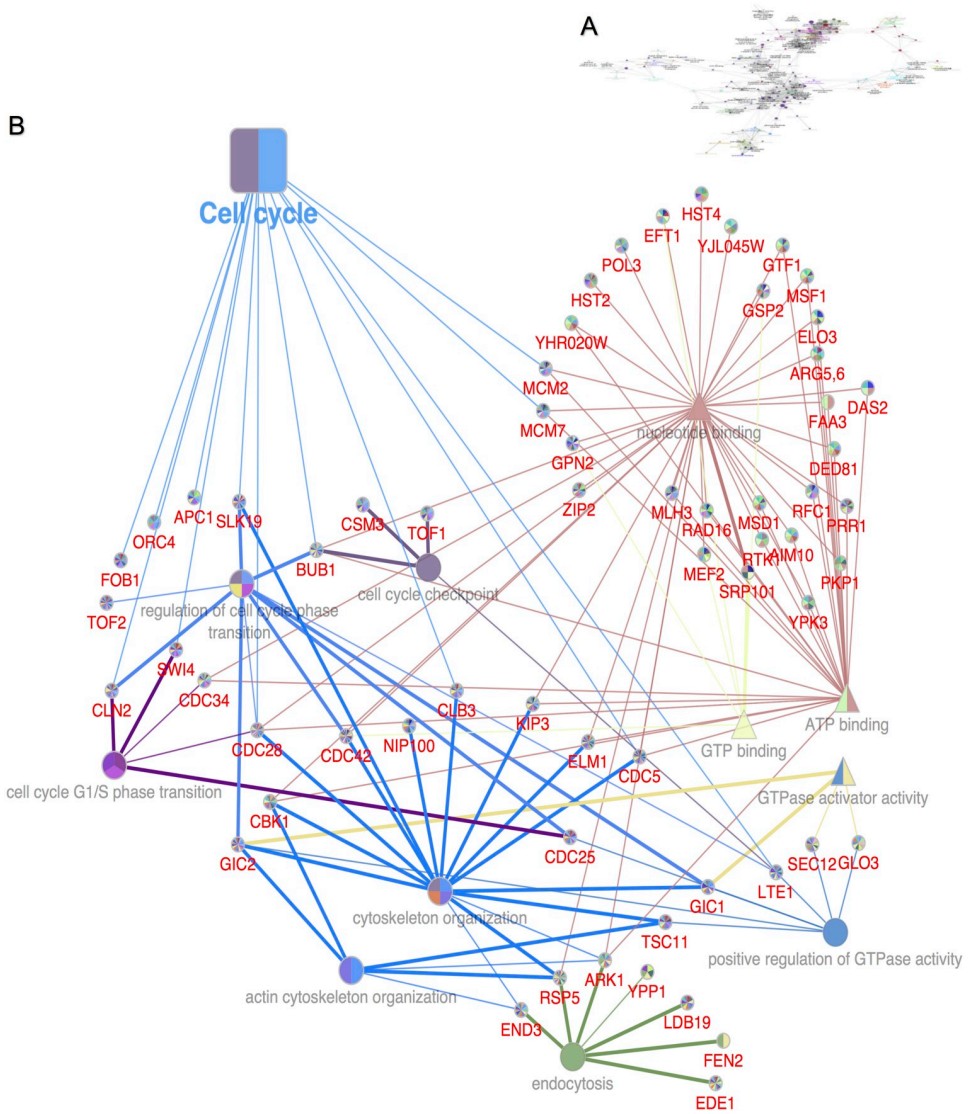

**Fig 6. Interactively functional annotation network of 238 differentially responsive mutants. A)** The functionally grouped network of KEGG pathway and gene ontology terms that are presented as nodes and linked to each other based on the similarity of their associated genes using ClueGo app along with Cytoscape software. **B)** Magnification of the highest significant node revealed the most significant node as the cell cycle pathway ($P \leq 0.001$). The square node represents the KEGG pathway, the circle nodes represent biological processes, and triangle nodes represent molecular function, while the colors denote clustered genes associated with the biological category and the size of node represents the $P$ values (0.05 as cutoff value). CluePedia app shows the genes for each node where the thickness of edges reflects the GO evidence code (thick line is based on experimental evidence whereas thin line is inferred from electronic annotation).

the highest significantly enriched pathway (cell cycle) to the less significant pathways (meiosis, DNA replication, endocytosis, and RNA biogenesis). When focusing on the cell cycle pathway only (Fig 6B), the non-essential genes (HOP profile) were strongly clustered with the essential genes (HIP profile) for significantly enriched biological processes associated with actin cytoskeleton organization and endocytosis. Eight clustered genes for actin cytoskeleton and endocytosis were found among the top 20 sensitive mutants of both the HIP and HOP profiles, implicating cell-cycle-dependent organization of the actin cytoskeleton and endocytosis as

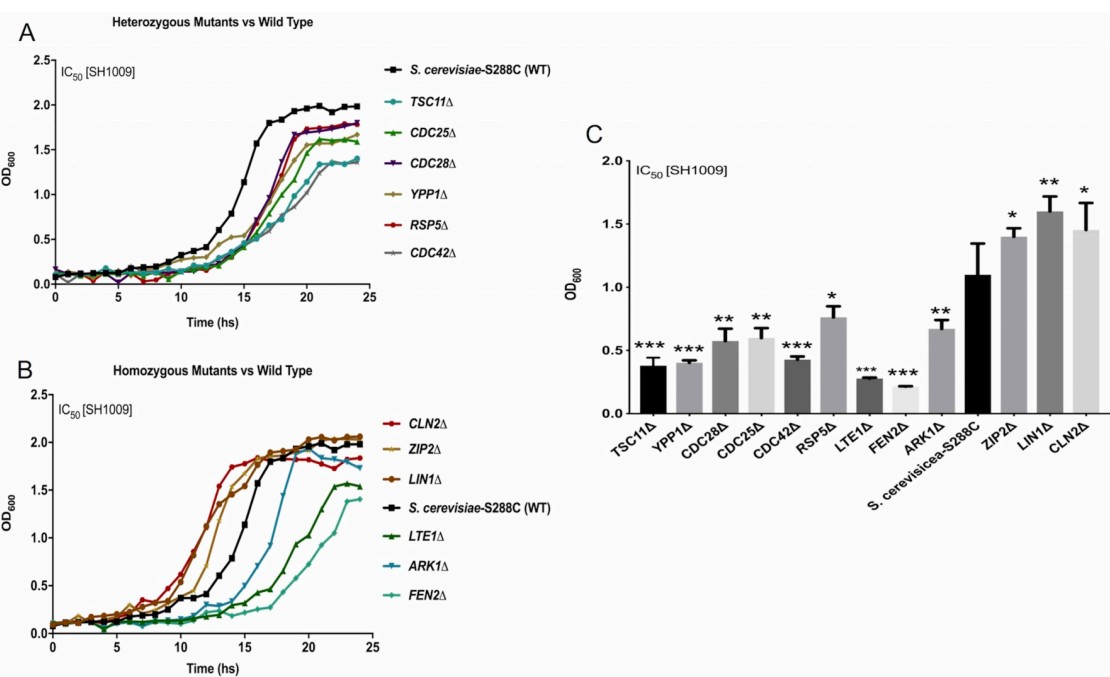

**Fig 7. Growth curves of SH1009-treated *S. cerevisiae*-S288C and mutants validate chemical-genetic interaction. A)** and **B)** The growth curves of the heterozygous mutants (HIP) and homozygous mutants (HOP), respectively, comparing to the wild type *S. cerevisiae*-S288C under the $IC_{50}$ concentration of aurone SH1009 treatment (16 μM) in YPD broth using the Bioscreen C growth curve instrument, and reading the $OD_{600}$ at 30 min intervals for 25 h during incubation at 30˚C. **C)** Individual significance comparisons of the growth of each mutant and wild type *S. cerevisiae*-S288C during exponential phase in the $IC_{50}$ concentration of aurone SH1009 are presented as mean±SEM using Dunnett's multiple comparisons test for heterozygous and homozygous mutants. *P* values (* $P \leq 0.05$) (** $P \leq 0.001$) (*** $P \leq 0.0001$).

targets of aurone SH1009. Additionally, the most significantly enriched molecular function for the SH1009-responsive mutants was the nucleotide-binding protein. The annotation network revealed 51 nucleotide-binding proteins that significantly responded to aurone SH1009; 31 of these genes encode proteins that act as ATP-binding proteins, while the other 10 encode GTP-binding proteins. Taken together, the functional enrichment analysis of the deletion pool of essential and non-essential genes illustrates that aurone SH1009 treatment might target nucleotide binding proteins, leading to a series of cellular defects that belong to cell cycle pathway, actin cytoskeleton organization, and endocytosis.

Before assessing biological responses to the aurone, growth curves were used to confirm the sensitivity or resistance responses of the individual mutants to SH1009 [32]. Twelve mutants that had more than or equal to a two-fold change and were also involved in the suggested biological responses from both HIP and HOP profiles were tested individually using dilutions of SH1009 (3.125–200 μM). The differences in the log-phase of the growth between the wild type *S. cerevisiae*-S288C and the mutants treated with SH1009 was compared. Fig 7 and S1 Fig depict the growth curves of individual *S. cerevisiae* deletion mutants for confirmation of the growth patterns of the HIP and HOP profiles.

The individual, heterozygous sensitive strains *TSC11Δ*, *YPP1Δ*, *CDC42Δ*, *CDC25Δ*, *CDC28Δ*, and *RSP5Δ* from chemical-genetic profiling all showed reduced growth with SH1009 treatment when compared to the log-phase growth of *S. cerevisiae*-S288C wild type, which supports results from chemical-genetic interaction analysis (Fig 7A). For the homozygous mutant strains, the same cellular effects were observed with decreased growth for sensitive mutants (*LTE1Δ*, *FEN2Δ*, and *ARK1Δ*) and increased growth for resistant mutants (*CLN2Δ*, *LIN1Δ*,

and *ZIP2Δ*) in the presence of SH1009 when compared to the wild type *S. cerevisie*-S288C (Fig 7B). Additionally, the growth of each mutant in the IC$_{50}$ concentration of SH1009 (16 μM) was compared independently to the growth of *S. cerevisiae*-S288C to assess significant differences between the growth of each mutant and the wild type. All mutants demonstrated significant differences in growth from the wild type ($P \leq 0.05$–0.0001), confirming their expected growth patterns as sensitive or resistant mutants to the aurone SH1009 and supporting the enrichment analysis results since these mutants harbor deletions of genes that are associated with cell cycle progression (*CDC25Δ, CDC28Δ, CLN2Δ, LIN1Δ,* and *ZIP2Δ*), actin cytoskeleton organization (*TSC11Δ, CDC42Δ, RSP5Δ, LTE1Δ,* and *ARK1Δ*), and endocytosis (*TSC11Δ, YPP1Δ, FEN2Δ,* and *RSP5Δ*) (Fig 7C). Once we confirmed the growth patterns of individual *S. cerevisiae* mutants to aurone SH1009, *C. albicans* was treated with the aurone to detect phenotypic changes associated with the suggested biological responses in the pathogen.

## Aurone SH1009 blocks cell cycle progression in *Candida albicans*

Cell cycle was the most significantly enriched pathway in the chemical-genetic interaction analysis (Figs 5 and 6). The other enriched pathways, meiosis and DNA replication, also contribute to the cell cycle pathway. Accordingly, if aurone SH1009 targets cell cycle gene-encoded proteins, the distribution of cell cycle phases in the fungal cell population during exponential phase should be altered compared to the normal distribution, indicating cell cycle arrest. For these experiments, changes in DNA content throughout different cell cycle phases was assessed by flow cytometry, which allows quantitative single cell detection. By labeling cellular DNA with propidium iodide (PI), the cells can be quantitatively discriminated in different phases of the cell cycle based on the fluorescence intensity, which is proportional to specific cell cycle phase [33]. Since G1 phase cells have a single copy of the genome, thus having the lowest amounts of DNA, whereas S phase cells are actively involved in DNA replication and will have increased amounts of DNA, and G2/M cells have two times the nuclear DNA of G1 phase cells, flow cytometric analyses of propidium iodide-stained nuclei can effectively differentiate G0/G1, S, and G2/M populations.

Before treating *C. albicans* SC5314 cells with SH1009 or cytochalasin D (CytoD) as a positive control for cell cycle arrest, a sample of early exponential culture was harvested and processed as described in the Materials and Methods to ensure cells were actively growing. The histogram in Fig 8A. indicates a rapidly dividing culture by having a significant fraction of cells (~70%) in the S phase. The histogram depicts an optimal flow-cytometric distribution for an actively growing yeast culture in rich media during early exponential phase, and was compatible with previous observations [33, 34]. After an additional 3 h of incubation without treatment, the untreated *C. albicans* SC5314 cells were still dividing with approximately similar cell cycle-distributed phases before three additional hours of incubation (Fig 8B). However, contrary to the cell cycle progression in untreated samples, the cell cycle distribution for *C. albicans* SC5314 cells that were incubated for 3 h with the SH1009 IC$_{50}$ concentration were distinctly perturbed (Fig 8C). There was a significant decrease in the fraction of cells in the S phase (46.88% compared to 70% in untreated cells, $P \leq 0.001$) and a significant increase in the proportion of cells in the G1 phase (46% compared to 16.53%, $P \leq 0.0001$), indicating accumulating cells in G1 phase (Fig 8E). As expected, 3 h of treatment with CytoD, an anticancer drug that inhibits the assembly and disassembly of actin subunits, led to a delay in the progression of G1 phase [35]. The DNA histogram revealed fewer cells in S phase and more in the G1 phase (52.68%, 39.17.0% and 2.13% in G0/G1, S and G2/M phases, respectively) (Fig 8D).

The accumulation of SH1009-treated cells in G1 phase implies that aurone SH1009 arrests cell cycle progression, supporting the chemical-genetic interaction results in which the most

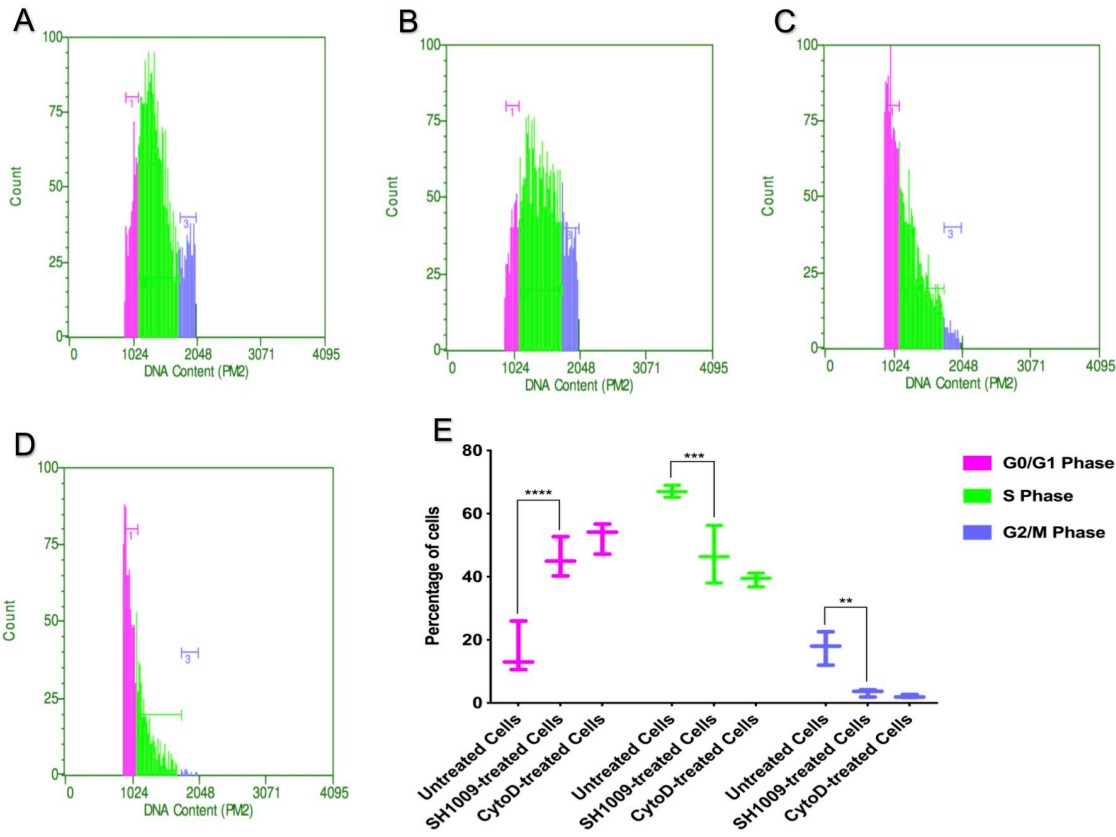

**Fig 8. Flow-cytometric analysis of the effects of aurone SH1009 on cell cycle progression in _C. albicans_ SC5314.** DNA histogram plots showing the percentages of cells in G0/G1 phase (pink peak on left), S phase (green center peak) and G2/M phase (blue peak on right) as a percentage of the total count of cells that were stained with Propidium Iodide (red fluorescence) and analyzed using the Millipore Guava flow cytometer and the Guava PCA-96 software system. A) Untreated cells at 0 time, B) untreated cells after 3 h of additional incubation, C) aurone SH1009-treated cells after 3 h of treatment with IC$_{50}$ SH1009 concentration (16 μM), and **D)** CytoD-treated cells after 3 h of treatment (25 μM) as a positive control. **E)** The significance comparison, _P_ values (**** $P \leq 0.0001$), (*** $P \leq 0.001$), (** $P \leq 0.01$) from a two-way ANOVA used to compare three population groups of G0/G1, S, and G2/M phases between untreated cells and SH1009-treated cells.

significantly responsive mutants to SH1009 essentially possess deletions for cell cycle-encoding proteins. The chemical-genetic analysis showed that homozygous deletion strain _CLN2_Δ, which lacks the G1 cyclin gene, was resistant to SH1009 (Fig 7B), and heterozygous deletion strain _CDC28_Δ, cyclin-dependent kinase (CDK), was sensitive to SH1009 (Fig 7A). In _S. cerevisiae_, cyclin Cln2 activates CdcC28 in late G1 phase, resulting in regulation of actin cytoskeleton polarization, which is crucial for bud emergence and G1 to S phase transition during cell cycle progression [36]. Therefore, we next sought to determine whether aurone SH1009 affected the disruption of the actin cytoskeleton.

## Aurone SH1009 perturbs actin cytoskeleton dynamics in _C. albicans_

The chemical-genetic interaction analysis indicated that cell-cycle-dependent organization of the actin cytoskeleton and endocytosis are targets of aurone SH1009 due to highly significant growth perturbation of 18 heterozygous and homozygous deletion mutants annotated as involved in actin cytoskeleton organization and endocytosis (Fig 6B). Fluorescent staining of actin and confocal laser scanning microscopy were used to visualize the effects of SH1009 on polarization of the actin cytoskeleton. SH1009-treated and untreated _C. albicans_ SC5314 cells

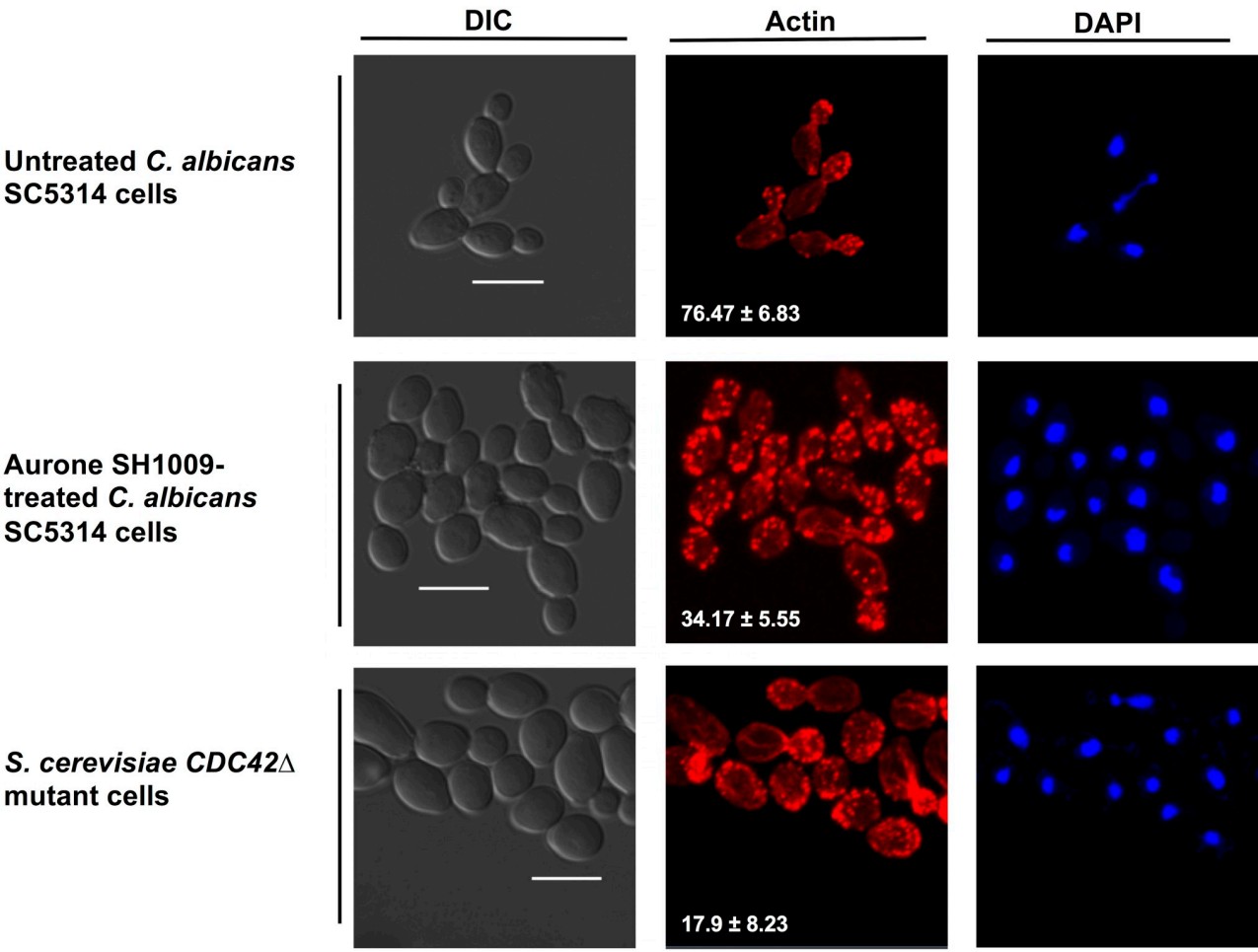

**Fig 9. Actin cytoskeleton dynamics in aurone SH1009-treated yeast cells.** *C. albicans* SC5314 and *S. cerevisiae CDC42Δ* mutant cells were fixed and the actin was stained with red-fluorescent rhodamine phalloidin (RP) and nuclear DNA was stained with blue-fluorescent DAPI. Scale bar = 5 μm. Quantitative data represent the percentage of cells that retained a polarized actin cytoskeleton as comparison between aurone SH1009-treated and untreated *C. albicans* SC5314 cells. Means±SEM (*P* value ≤0.01).

along with the *S. cerevisiae CDC42Δ* heterozygous deletion strain with a deletion for an actin-regulatory gene were examined microscopically for phenotypes indicative of disruption of actin distribution. In late G1 phase of the cell cycle of budding yeast, the actin patches should normally be assembled in the tip of the small-growing bud with clear actin cables that are polarized from the bud tip towards the mother cell. Actin cables serve as tracks to transport mRNAs, proteins, mitochondria, and ribosomes from the mother cell to the daughter cell. Actin patches are clustered in the secretion and endocytosis sites for critical roles in plasma membrane invagination during endocytosis [37] and cell wall remodeling through orienting the polarized secretion of cell wall constituents and enzymes towards the actin patches, promoting bud emergence [38, 39].

The images in Fig 9 depict typical actin polarization for untreated *C. albicans* SC5314 cells during exponential phase in which ~76.47% of cells were observed with red-fluorescent actin patches concentrated in the buds along with polarized-apical cell growth. However, after 3 h of SH1009 treatment, *C. albicans* cells appeared abnormally round and enlarged compared to untreated cells, indicating depolarized growth (Fig 9, DIC panel). In addition, only ~34.17% of

SH1009-treated cells retained actin patches in the buds while the remaining had actin patches that were scattered randomly in both the mother cell and buds, signifying actin depolarization. In addition, SH1009-treated cells displayed a distinctively distorted actin assembly resulting in a considerable number of large aggregates of the actin (Fig 9, Actin panel). This phenotype was previously associated with endocytic mutants such as, but not limited to, *ARK1Δ* [40], *END3Δ* [41], and *RPS5Δ* [42]. In growth curve studies with SH1009-individually treated deletion mutants, these strains were either sensitive (*ARK1Δ* and *RPS5Δ*) or resistant (*END3Δ*) to SH1009 (Fig 7A and 7B) and were also identified in our chemical-genetic interaction analysis (Fig 6B). A previous study that investigated the composition of the actin clumps using immune-electron microscopy detected an accumulation of endocytic vesicles surrounded with actin filaments and a mixture of actin patches and endocytic proteins, suggesting a failure to mature the endocytic vesicles properly as a consequence of the inability to disassemble the actin-associated endocytic complexes [43].

The actin distribution in the *S. cerevisiae CDC42Δ* deletion mutant was also investigated due to its sensitivity to SH1009 (Fig 7A). In the majority of *S. cerevisiae CDC42Δ* mutant cells, there was an accumulation of large, round, unbudded cells with distributed actin patches (Fig 9, DIC panel), which supports previous reports [36]. The depolarized growth as well as the abnormal distribution of the actin patches in SH1009-treated *C. albicans* cells support inhibition of cell cycle arrest by preventing G1 phase progression (Fig 8D). In yeast cells, it has been evidenced in several studies that failure in rearrangement of actin patches at the bud site in G1 phase will prevent the emergence of the bud and arrest the cell cycle at G1 phase [44, 45]. These findings are compatible with previous observations that have established a definitive link between intact actin cytoskeleton organization and cell cycle progression [35].

## Aurone SH1009 alters *C. albicans* expression of genes involved in the cell cycle, actin polarization, and endocytosis

To study changes in gene expression in response to aurone SH1009, quantitative reverse transcription-PCR was employed to confirm mRNA abundance of a set of *C. albicans* SC5314 genes homologous to the *S. cerevisiae*-S288C genes in the heterozygous and homozygous deletion mutants identified as significantly enriched in pathways from the chemical-genetic interaction analysis (Fig 5). The cell cycle-associated genes were the most responsive to aurone SH1009. Successful cell cycle progression requires expression of checkpoint genes that guarantee sequential execution of certain cellular processes. For instance, before chromosome segregation, chromosomal DNA must first be replicated. These processes are ordered by activation and inactivation of (CDK) Cdc28, an ATP-binding protein, which complexes with activating subunits called cyclins [44]. *HGC1*, is a *C. albicans* gene homologous to *S. cerevisiae* G1 cyclin gene *CLN2*, encoding a G1 cyclin that binds with Cdc28 to form a cyclin-CDK complex with a central role in G1/S transition during the cell cycle [46, 47]. SH1009-treated *C. albicans* SC5314 results in a ~2.3 and 4.9-fold respective downregulation of *CDC28* and *HGC1* gene expression after 1.5 h of treatment (Fig 10), supporting a role for SH1009 in arresting the *C. albicans* cell cycle in G1 phase (Fig 8C).

Binding of Cdc28 to Hgc1 is an important regulator step for controlling cell cycle progression. This complex regulates polarity by phosphorylating and blocking Rga2 (a negative regulator of Cdc42), thus activating Cdc42 (GTP-binding protein), leading to sustained actin polarization and hyphal growth [48]. Failure in the relocation of Cdc42 by the Cdc28-Hgc1 complex to either the bud site or hyphal tip in late G1 phase results in a haphazard distribution of actin cytoskeleton and subsequently accumulation of unbudded, enlarged cells [36, 45]. The *S. cerevisiae CDC42Δ* deletion mutant was hypersensitive to aurone SH1009 treatment (>70%

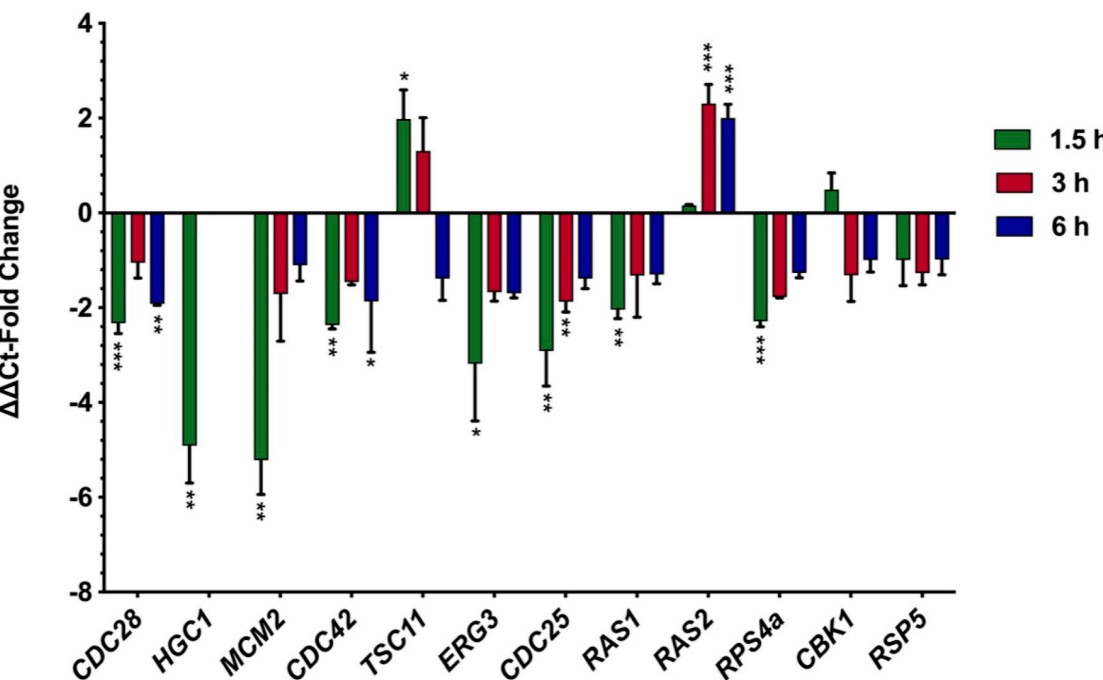

**Fig 10. Expression of genes in aurone SH1009-treated *C. albicans* SC5314.** The relative gene expression of normalized transcript levels of treated samples was calculated by comparison to normalized transcript levels of untreated samples ($2^{-\Delta\Delta Ct}$) for cell cycle- and actin cytoskeleton-associated genes at indicated time points of treatment. The data are means from two biological replicates, each with three technical replicates. The fold changes that are $\geq 1.8$ with $P$ values $\leq 0.05$ (* $P \leq 0.05$), (** $P \leq 0.001$), (*** $P \leq 0.0001$) were considered significant for upregulation (positive scores) or downregulation (negative scores).

inhibition) at a concentration of 16 μM (Fig 7A). In SH1009-treated *C. albicans* SC5314, *CDC42* expression was downregulated two-fold (Fig 10), which would confirm the downstream effect of SH1009 on expression of *CDC28* and *HGC1*.

Cdc28 also controls the cell cycle by regulation of DNA replication. The *MCM2* gene encodes an ATP-binding protein that is a part of the pre-replicative complex known as minichromosome maintenance, consisting of proteins Mcm2-7, which acts as a helicase to unwind DNA and initiate DNA replication [49]. In *S. cerevisiae*, after assembly of the MCM2-7 complex in G1 phase, MCM2-7 requires phosphorylation at the end of G1 by a Cdc28-kinase complex in order to recruit DNA polymerase and initiate DNA replication [50]. SH1009 treatment resulted in a significant five-fold decrease in MCM2 by 1.5 h after treatment (Fig 10), which could represent an additional downstream effect of SH1009 on Cdc28. Mmc2 is also an ATP-binding protein and, along with 12 other proteins enriched by chemical-genetic interaction analysis, contains the widely-distributed P-loop motif, supporting nucleotide-binding proteins as potential targets for SH1009 [51].

In addition to *CDC28*, other genes that encode for nucleotide-binding proteins with roles in signal transduction pathways were identified in deletion mutants with differential sensitivities to aurone SH1009. The *C. albicans CDC25* and *RAS1* genes were downregulated by an approximate three-fold and two-fold, respectively (Fig 10). *CDC25* encodes a GTP-binding protein that acts as a guanine nucleotide exchange factor and is the upstream activator of the Ras/cAMP signaling pathway, responsible for activating the GTP-binding protein, Ras1, which then activates the cAMP synthesis required for cell cycle progression [52, 53]. Additionally, Ras1 activation of cAMP synthesis leads to maintenance of the hyphal growth, a virulence factor in systemic candidiasis, by regulating the G1 cyclin Hgc1 which binds to Cdc28 to

localize Cdc42 to the hyphal tip [46]. As evidenced by a previous study, SH1009 was the only aurone compound that disrupted the biofilm formation [21], which would require inhibiting hyphal growth as a key component of biofilm formation.

In contrast to decreased expression of *RAS1* transcripts, *RAS2* transcripts were upregulated two-fold after 3 h of SH1009 treatment (Fig 10). *RAS2* encodes a GTPase in *C. albicans*, and a previous study has shown cAMP levels in a *RAS1Δ* mutant declined 20-fold yet increased by ~10% in a *RAS2Δ* mutant, indicating an antagonistic activity of *C. albicans* Ras2 on the cAMP levels through an unknown mechanism. In *S. cerevisiae*, both Ras1 and Ras2 are required for activating cAMP synthesis. However, unlike in *S. cerevisiae* in which Ras1 and Ras2 have been well studied and share sequence homology, *C. albicans RAS2* has poor sequence homology with *S. cerevisiae RAS* sequences [54]. Although the response of *RAS2* to SH1009 is unclear due to the lack of studies on *C. albicans* Ras2 activity or its interaction with other members of the Ras/cAMP signaling pathway, the upregulation of transcripts may be a response to down-regulation of upstream molecules affected by SH1009.

Another transcriptional upregulation in SH1009-treated *C. albicans* was increased expression of *TSC11* after 1.5 h of aurone treatment (Fig 10). *TSC11* encodes a GTP-binding protein characterized as regulating actin cytoskeletal dynamics during polarized growth and endocytosis [55, 56]. The increased expression of *TSC11* could be a detoxification mechanism for *C. albicans* to counter the negative effects of SH1009 on actin cytoskeletal dynamics and endocytosis. Previous studies reported phenotypes of *TSC11Δ* mutants (also known as *AVO3* or *RICTOR* in mammals) as having abnormal actin polarization [57], reduced endocytosis rate [58], and arrested cell cycle progression [59]. Another gene involved in endocytosis exhibiting altered expression in response to SH1009 was *ERG3* which was downregulated three-fold (Fig 10). *ERG3* is involved in the ergosterol biosynthesis pathway, and gene deletion has been documented as attenuating the endocytosis rate [60]. Reduced expression of *ERG3* could also be attributed to the upstream effects of SH1009 on the RAS/cAMP signaling pathway that positively regulates the expression of the *ERG* gene family [61].

Gene expression of *RPS4a*, which encodes a 40S ribosomal protein, decreased two-fold with SH1009 treatment (Fig 10). A genome-wide study of haploinsufficient *C. alibcans* deletion mutants that were fractionated for abnormal cell size revealed genes associated with ribosome biogenesis and cell cycle, suggesting a correlation between ribosome biogenesis rate and size-dependent cell cycle progression [62]. In *S. cerevisiae*, repressing the synthesis of this conserved gene, *RPS4a*, results in arresting G1 phase or a significantly prolonged G1 phase, along with the phenotype of increased cell size [63], which could explain the enlarged cell phenotype observed microscopically in SH1009-treated *C. albicans* cells (Fig 9).

## Discussion

In this study, we investigated *in vitro* antifungal activity of the bioactive compound, aurone SH1009, against widely used CLSI reference resistance tester strains and additional strains of *Candida* spp. using the CLSI broth microdilution method [22]. Aurone SH1009 was not only inhibitory for susceptible strains, but was also inhibitory for resistant-clinical isolates. The clinical isolates, *C. albicans* M4: Gu2 and *C. albicans* M6: F1 were isolated initially as fluconazole-susceptible strains from two HIV-infected patients who were suffering from recurrent oropharyngeal candidiasis. After a two-year period of fluconazole treatment, the last isolates in each series were fluconazole-resistant counterparts known as *C. albicans* M5: Gu5 and *C. albicans* M7: F5 [64]. *C. albicans* M5 and M7 have acquired gain of function mutations, leading to homozygous alleles for the transcriptional factors *MRR1* and *TAC1* which have been documented for overexpressing the *CDR1/2* (Candida Drug Resistance) and *MDR1* (Multi-Drug

Resistance) genes for azole-efflux pumps, respectively [65]. Additionally, *C. albicans* M2: ScTAC1R34A and *C. albicans* M3: ScMRR1R34A are fluconazole-resistant strains that have been mutated in *C. albicans* SC5314 background to encode $MRR1^{P683S}$ and $TAC1^{G980E}$ homozygous activating alleles, respectively. As a consequence of these mutations, the expression of efflux pumps are induced, causing a reduction in intracellular accumulation of azoles and ultimately high fluconazole resistance [66, 67].

Loss-of-heterozygosity events as genetic alterations are commonly associated with fluconazole resistance, reflecting the capacity of *C. albicans* to generate adaptive-homozygous mutations, which is attributed to the extraordinary plasticity of the *C. albicans* genome [68, 69]. Aurone SH1009 exhibited considerably low $IC_{50}$ values in the inhibition of these fluconazole-resistant isolates, indicating that the highly expressed efflux pumps could be modulated by aurone SH1009. The efflux pumps *CDR1/2* are mainly ATP-binding cassette (ABC) transporter proteins that harness the energy from ATP hydrolysis in order to extrude drug out of the yeast cell [7]. In fact, as consistent with previous studies on cancer cells, aurones selectively inhibited the pumping action of these transporters by binding with the C-terminal nucleotide-binding domain of P-glycoprotein which belongs to the ATP-binding cassette (ABC) superfamily [70, 71]. The modulatory effect of aurones on drug-efflux pumps allows the intracellular accumulation of the compound and its associated toxicity, which reverses the resistance mechanism and renders the resistant cells as susceptible again, making these bioactive compounds attractive candidates as reversal agents to control drug resistance [72].

Multidrug resistant isolate *C. albicans* ATCC 64124 is known as Darlington's strain, the name of the patient suffering from chronic mucocutaneous candidiasis [73]. Darlington strain, 64124, is defined as resistant to all antifungal agents except 5-fluorocytosine at high concentration [74]. The resistance mechanisms toward multiple classes of antifungal agents of Darlington's strain have not been studied, but evidence of the evolution of multidrug resistance has been published recently, demonstrating the capacity of clinical *C. albicans* isolates to adapt during long-term antifungal treatment [75]. Multidrug resistance is recognized as a multistep process resulting from a gradual accumulation of mutations during long-term antifungal treatment. Briefly, the resistance began with fluconazole treatment that causes an activation mutation in *TAC1*, followed by caspofungin exposure that leads to a mutation in *FKS1*. Then, amphotericin B treatment causes a loss of function mutation in *ERG2*. Consequently, the final strain was a multidrug resistant strain that acquired all the three mutations and exhibited resistance to three classes of the standard antifungal drugs [7].

Unlike *C. albicans* that acquires the resistance, *C. glabrata* is resistant to azole drugs due to ~67 gain of function mutations in the transcription factor *CgPDR1* gene, resulting in overexpression of *CgCDR1/2*-encoded efflux pumps [76]. This could account for the dramatic increase in *C. glabrata* infections since the introduction of azole drugs in the 1980s, identifying *C. glabrata* as the second most common non-*albicans* cause of candidiasis. Drug resistance of *C. glabrata* has not only been found for azoles, but also for all known antifungal drugs, posing a serious challenge to antifungal therapy for the organism [77]. Our results indicated that aurone SH1009 effectively inhibited *Candida* spp. growth and exhibited acceptable pharmacodynamics properties (time- and concentration-dependent killing) that were similar to fluconazole, as a valuable estimation for a prospective antimicrobial agent [78].

After establishing SH1009 antifungal activity, we proceeded to characterize the antifungal mechanism. Studying the mode of action of a new drug is a critical prerequisite to improve therapeutic action and avoid unwanted side effects, but it has been a challenge in drug discovery due to the high consumption of cost and time. However, since the first yeast deletion collection was created in 2002 [79], chemogenomic profiling has been proven as a powerful system for characterizing the mode of action and molecular targets of several drugs [80], as

well as thousands of additional bioactive molecules [81, 82]. A considerable number of studies have been recently reviewed the broad spectrum of biological activities of aurone compounds as anticancer agents identified using phenotype-based approaches, exploiting the existing knowledge of a given cellular process inhibited by aurones, including inhibition of CDKs, DNA scissoring, histone deacetylase, topoisomerase, ATP-binding cassette transporter, and tubulin polymerization [83]. In contrast, here, a genetic-based approach was applied to comprehensively reveal all the biological effects of aurone treatment as a potential antimicrobial. The chemogenomic approach that was used in this study is a reverse-genetics technique that allows for comprehensive identification of gene products that functionally interact with bioactive molecules by exposing previously constructed libraries of *S. cerevisiae* mutants to the condition of interest, which in this study was exposure to the novel aurone SH1009. These yeast mutants harbor deletions of the majority (96%) of the yeast genome with only one mutant per deletion-gene. Due to nearly complete coverage of the yeast genome, this approach guarantees unbiased results because it completely maps all targets (biological responses) simultaneously under the term of interconnected pathways of eukaryotic cell [84].

In this study, chemogenomic profiling of *S. cerevisiae* mutants with aurone SH1009 identified roles for genes involved in cell cycle, DNA replication, cell division, actin cytoskeleton, and endocytosis. Enrichment analysis for biological responses in heterozygous and homozygous mutants was first performed separately to confirm significant responses in each pool. The responses were then combined to map the functional annotation network of KEGG pathways, biological responses, and molecular functions simultaneously using GO enrichment analysis to find that the top responsive heterozygous and homozygous mutants clustered for actin cytoskeleton and endocytosis processes. GO enrichment analysis was also used to map the cell cycle pathway as the most significantly enriched pathway, also encompassing the other enriched biological processes and revealing nucleotide-binding proteins as the molecular function of the most of significantly enriched genes.

Nucleotide-binding proteins play a central role in a variety of pivotal cellular processes, including cell signaling, proliferation, cytoskeletal assembly, protein synthesis, and apoptosis [85]. The low number of GTP-binding proteins that were identified in this study could be attributed to a relatively low abundance of GTP-binding proteins in nature compared to ATP-binding proteins [86]. Nevertheless, because purine nucleoside triphosphates share a similar structure, some ATP-binding proteins may utilize GTP as the phosphate donor and vice versa [87]. In this respect, aurone SH1009 might possess an affinity for the same binding site in different but related nucleotide-binding proteins, hence, affecting a broader spectrum of cellular processes. The functional enrichment of genes encoding nucleotide-binding proteins during analysis of the deletion pool of essential and non-essential genes suggests that aurone SH1009 potentially interacts with nucleotide-binding proteins, leading to a series of cellular defects impacting the cell cycle, actin cytoskeleton, and endocytosis. The benzofuranone ring of SH1009 may dictate the biological activity due to mimicry of adenine, subsequently, inhibiting the activity of ATP-dependent proteins [88].

Our chemical genetic interaction and enrichment analysis also suggest that treatment with aurone SH1009 targets nucleotide-binding proteins, producing alterations in cell cycle, actin cytoskeleton organization, and endocytosis in *S. cerevisiae*. These phenotypic changes in response to SH1009 were confirmed in *C. albicans* SC5314 using cell sorting analysis by flow cytometry, which delineated increased populations of cells in G1 phase after 3 h of aurone SH1009 treatment. In *S. cerevisiae*, heterozygous mutants with gene deletions for proteins affecting the cell division cycle (*CDC5Δ*, *CDC34Δ*, *CDC42Δ*, *CDC25Δ*, and *CDC13Δ*) that sensitively responded to aurone SH1009 are all controlled by the master regulator of the cell cycle, (CDK) Cdc28, a catalytic kinase that couples with activating subunits such as Clb3 and Cln2

[89], the genes for which both responded to SH1009 as homozygous-resistant mutants. Also, the accumulation of SH1009-treated *C. albicans* cells in G1 phase implies that the cells are in cell cycle arrest and do not yet engage in DNA synthesis, also supporting the chemical-genetic interaction results in which DNA replication mutants *POL3Δ*, *MCM2Δ*, *MCM7Δ*, *RFC1Δ*, *RFC3Δ*, and *ORC4Δ* from the HIP profile were sensitive to SH1009. In *C. albicans*, gene expression during different cell cycle stages has been previously characterized using microarray technology [90]. This study reported expression of several genes that are required for G1/S transition and some of these genes are identified as sensitive or resistant mutants from our chemical-genetic profiles, including *POL3* for DNA polymerase subunit, *RFC1* and *RFC2* for DNA replication factor elements, *TOF1* gene for a DNA replication checkpoint, *CSM3* gene for accurate chromosome segregation, *SWI4* transcription factor for activation of G1 transition (Fig 6).

Our results are also in agreement with a prior study reporting that aurones arrest the cell cycle via targeting (CDK) protein Cdc28, making aurones a promising candidate for cancer therapy [12]. Among different aurone derivatives synthesized as potential anticancer agents, aurone compounds that possess a methoxy group at position 4 on ring B, like aurone SH1009, were found to produce an enhanced antiproliferative activity by significantly arresting the cell cycle at G2/M phase [83]. In our study, fluorescent imaging was performed to detect actin cytoskeleton depolarization as a consequence of aurone SH1009 perturbation of rearrangement of actin patches in the bud site at the end of G1 phase, leading to cell cycle arrest at G1 phase. These phenotypic findings support the chemogenomic responses of *S. cerevisiae* genes *CDC28Δ*, *CLN2Δ*, and *CDC42Δ* to SH1009, because it has been well characterized that the G1 cyclin Cln2 binds to Cdc28 to localize Cdc42 to promote actin polarization and G1/S phase transition [36].

Chemogenomic profiling does have limitations; for instance, cellular mechanisms affected by SH1009 were identified using growth as a sole endpoint measurement, potentially resulting in selection for false positive sensitivities due to slow growth of some deletion mutants or, in the worst scenario, loss of indispensable mutants. Another limitation is that responses in *S. cerevisiae* differ from responses in *C. albicans* because of the ability of *C. albicans* to produce true hyphae and differences in sensitivity and resistance to antifungals. To confirm specific gene responses in *C. albicans*, relative levels of mRNA of genes identified as differentially responsive were quantified in *C. albicans* SC5314. A sizable subset of genes that function in cell cycle progression was differentially regulated in response to SH1009, leading to a potentially hierarchal pathway beginning with Cdc25 and Ras1 that regulate G1 cyclin Hgc1, which binds to Cdc28 in order to localize Cdc42, leading to actin polarization and cell cycle progression. Cdc25 and Ras1 regulate the RAS/cAMP signaling pathway which controls Erg3 maintenance of normal endocytosis rates. The transcriptional changes in the genes for these molecules support the biological responses observed in phenotypic studies.

Two genes, *CBK1* and *RSP5*, identified by chemogenomic profiling in *S. cerevisiae*, did not demonstrate significant differential expression to SH1009 treatment in *C. albicans* in spite of their respective documented roles in polarized growth [91] and organized actin patches with normal endocytosis [42]. *CBK1Δ* mutant was among the top 20 sensitive heterozygous deletion mutants (S2 Table), and the *RSP5Δ* mutant was confirmed using growth curves for its sensitivity to SH1009 (Fig 7A). Failure to detect changes in gene expression could be related to the timing of expression or discordance between regulation and gene expression. In *S. cerevisiae*, the *TSC11Δ* mutant was hypersensitive to SH1009, however, this gene was transcriptionally upregulated in *C albicans*, which could indicate a difference in survival responses between the two yeasts. This highlights that although this study has validated chemical-genetic interaction using *S. cerevisiae* mutant collections to characterize the mode of action for a novel bioactive

aurone compound against *C. albicans*, chemical-genetic interaction studies could also be used to elucidate potential differences in drug responses between organisms.

## Conclusion

In summary, we investigated aurone SH1009 for antifungal activity, human cell line cytotoxicity, and pharmacodynamics properties. Aurone SH1009 exhibited promising selectively fungistatic-inhibitory activity against *Candida* spp., particularly resistant isolates, highlighting the possibility of a broad-spectrum property for aurone SH1009 against other pathogenic fungi. We also provided the first, comprehensive genome-wide study of haploinsufficiency and homozygous screening in *S. cerevisiae* for a bioactive aurone compound. Chemical-genetic interaction analysis predicted that SH1009 targets cell cycle-dependent organization of the actin cytoskeleton and endocytosis, suggesting a novel mode of action for this aurone compound. Phenotypic studies for these significantly enriched biological responses were completed in *C. albicans* demonstrating G1 phase-arrested cells with abnormal, depolarized actin cytoskeleton. Differential expression of genes, identified by chemical-genetic interaction, in response of SH1009 in *C. albicans* coincided with enrichment of cells in G1 phase and actin depolarization producing a model that offers a significant improvement in understanding the mechanism for toxicity. The current study provides experimental frameworks for future mechanistic studies that could be used to investigate the bioactivity of a large number of aurone compounds using chemical-genetic interaction and phenotypic-based methods.

## Materials and methods

### Materials and reagents

*Candida* strains (listed in S1 Table) were provided by the Dr. P. David Rogers lab of the University of Tennessee Health Science Center, Memphis, TN. Human cell lines; THP-1 (ATCC, TIB-202), HepG2 (ATCC, HB-8065), and A549 (ATCC, CCL-185) were purchased from the American Type Culture Collection (Manassas, VA, USA). Dulbecco′s Modified Eagle′s culture medium (DMEM), 1X trypsin-EDTA solution, fetal bovine serum (FBS), 100X penicillin/streptomycin solution, Amphotericin B, caspofungin, fluconazole, itraconazole, 5-fluorocytosine, YPD agar and broth, cytochalasin D, 3-(N-morpholino) propanesulfonic acid (MOPS) buffer, and methyl methanesulfonate (MMS) were all purchased form Sigma-Aldrich (St. Louis, MO, USA). RPMI-1640 medium was purchased from Corning Incorporated (Coring, NY, USA). Geneticin selective antibiotic (G-148 sulfate), phosphate buffered saline (PBS), rhodamine phalloidin, propidium iodide, PrestoBlue, and RNaseA enzyme were purchased from Life Technologies Corporation (Carlsbad, CA, USA). 4% Paraformaldehyde was purchased from Alfa Aesar (Ward Hill, MA, USA) and zymolase 20T was purchased from MP Biomedicals, LLC (Solon, OH, USA). The yeast deletion collections (~1,056 heterozygous mutants and ~4,320 homozygous mutants) were purchased from GE Healthcare Life Sciences (Pittsburg, PA, USA) and ThermoFisher Scientific (Waltham, MA, USA), respectively.

### Antifungal susceptibility testing

**Preparation of stock solutions.** Aurone SH1009 was synthesized as described in supporting information (S1 File). The powder of SH1009 was dissolved in dimethyl sulfoxide (DMSO) to a high concentration of 20 mM. Two-fold serial dilutions were prepared at concentrations from 200 to 3.125 μM using RPMI–1640 medium that was buffered previously to pH 7.0 with MOPS and sterilized by filtration. Using 96-wells microtiter plates, 100 μL of each SH1009 concentration was added to respective wells with four replicates for each concentration.

Amphotericin B and caspofungin were used as positive controls to ensure 100% growth inhibition of the yeast at concentrations of 16 μg/mL and 8 μg/mL, respectively. In addition, fluconazole, itraconasole, and 5-fluorocytosine were prepared following manufacturer instructions, serially diluted two-fold according to CLSI protocol concentrations [22], and used as a reference for some of the strains to confirm their resistance profiles.

**Preparation of inocula.** Strains for this study (S1 Table) were cultured on YPD agar and incubated at 35˚C for 24 h. The inoculum suspensions for each strain was prepared according to the CLSI broth microdilution protocol for antifungal susceptibility testing of yeasts [22]. The suspension of 5–6 colonies was vortexed in approximately 4 mL of sterile saline solution (0.85% NaCl) and adjusted spectrophotometrically to optical densities at a 530 nm wavelength ($OD_{530}$) that ranged from 0.12 to 0.15. The inocula were then diluted 1:1000 in RPMI 1640 medium resulting in $1 \times 10^3$ to $5 \times 10^3$ CFU/mL working concentration. Volumes of 100 μL of each inoculum for each strain were added to the wells of its respective plate. For each isolate, there were drug-free wells and media-control wells with and without 1% DMSO to detect any contamination in the media and for use as an optical blank for optical density and fluorescence measurements.

**Inhibition assay.** After 24 h of incubation at 35˚C, 20 μL of PrestoBlue reagent were added to each microtiter well to a final concentration of 10% after which plates were incubated at 35˚C for an additional 60–70 min [21]. The minimum inhibitory concentrations (MICs) were defined as the concentration of aurone 1009 that reduced the growth by 90%. The MIC value was determined quantitatively by measuring the fluorescence that results from reducing blue-nonfluorescent resazurin, to red-fluorescent resorufin as a result of metabolic activity of the active cells at 560 nm excitation and 590 nm emission with a SpectraMax M5e spectrophotometer (Molecular Devices, LLC, USA). The percentages of yeast growth were calculated by comparing the fluorescent readings of the drug-containing wells with that of the drug-free wells to calculate $MIC_{90}$. The assay for each strain was performed in duplicate. The $IC_{50}$ values were calculated using GraphPad Prism (GraphPad Software, USA).

**Cell viability assay.** Using 24-well plates, *C. albicans* SC5314 cells were cultured and treated with 200 μM SH1009 in 2 mL of RPMI 1640 medium as described above. After incubation at 35˚C for 48 h, 1 mL of SH1009-treated cells, untreated cells, and previously prepared isopropanol-killed cells were washed with PBS. The samples were then diluted to approximately $1 \times 10^6$ cells/mL in PBS and stained according the live/dead Fungalight Yeast Viability Kit protocol (ThermoFisher, Waltham, MA, USA). Stained cells were investigated for their viability using flow cytometry (Guava Millipore, Burlington, MA, USA) with Guava InCyte Software. To distinguish between live and dead cells in a dot plot, gating the yeast population was determined based on red and green fluorescence levels of isopropanol-killed cells.

**Growth rate assay.** In 100-well Bioscreen honeycomb plates, *C. albicans* M1:SC5314 cells were cultured and treated with aurone SH1009 (3.125–200 μM) in RPMI 1640 medium as described above. Plates were loaded into the Bioscreen C instrument with Bioscreen software (Growth Curves USA, Piscataway, NJ, USA) at a temperature of 35˚C with continuous shaking and 30 min interval measurements at an $OD_{530}$ for 40 h. Growth curves were used to compare growth of untreated-*C. albicans* cells with SH1009-treated cells at different aurone concentrations. Validation of cellular sensitivity and resistance responses of *S. cerevisiae* mutants to SH1009 were also performed with the Bioscreen C instrument as the same manner for *C. albicans* except the media was YPD broth and incubation was at 30˚C.

**Time-kill assay.** A previously described and evaluated antifungal time-kill method was utilized to evaluate the fungicidal activity of aurone SH1009 [92]. An initial inoculum of *C. albicans* SC5314 ranging from 0.5–1.0 $\times 10^5$ CFU/mL was treated with an aurone concentration approximately five-fold higher (500 μM) than the dilution producing the $IC_{50}$ of SH1009

(16.25 μM) in the two-fold dilution series. Fluconazole (16 μg/mL) was used as fungistatic control, and Amphotericin B (8 μg/mL) was used as fungicidal control. After incubation at 35˚C, 10 μL of each treatment was spread onto YPD agar after 0, 6, 12, 24, and 30 h of treatment and plates were incubated at 35˚C for 24 h to determine viable cell numbers. The fungicidal activity was determined as $\geq 3$ –$\log_{10}$ which is equivalent to 99.9% reduction in CFU/mL from the working concentration 0.25–0.5×$10^5$ CFU/mL.

## Cytotoxicity assay

The A549 human lung carcinoma epithelial cell line and human monocytic THP-1 cell line were grown in RPMI-1640 culture medium, while HepG2 human liver carcinoma epithelial cells were grown in DMEM culture medium. Both media were supplemented with 10% FBS and 1% penicillin-streptomycin antibiotics. After maintaining the cell growth at 37˚C with 5% $CO_2$ in a humidified incubator until reaching 90% confluency, the A549 and HepG2 cells were trypsinized with 1X trypsin-EDTA and resuspended in fresh medium. The cells were seeded into 96-well microtiter plates at a density of 10,000 viable cells/well and grown overnight, while the suspension THP-1 cells were seeded directly into 96-well microtiter plates at the same density before treatment. The final concentrations of aurone SH1009 were prepared in two-fold serial dilutions (3.125 μM– 200 μM) as described above for the antifungal susceptibility assay. The media containing A549 or HepG2 cells were then replaced after overnight incubation by 200μL of fresh culture media containing the final concentrations of SH1009. The cells were then incubated for additional 24h at 37˚C with 5% $CO_2$ in a humidified incubator. To evaluate cell viability, each well was treated with 20 μL of PrestoBlue for 3–6h. Metabolically active cells converted the blue non-fluorescent dye resazurin to the pink fluorescent dye resorufin, which can be measured by plate reader as described above in antifungal inhibition assay. Triton X-100 (1%, *v/v*) was used as a positive control to give a complete loss of cell viability. Percentages of cell viability were calculated as follows: [(negative control value–treated value) × 100]/negative control value. The assay for each cell line was performed in triplicate. The $CC_{50}$ values were calculated using GraphPad Prism (GraphPad Software, USA).

## Chemogenomic profiling in *Saccharomyces cerevisiae*

**Combination of individual mutants into a single pool.** The yeast deletion collections were obtained as individual mutants in 96-well plates that had been stored at −80˚C. Both the HIP deletion pool and HOP deletion pool were created separately as previously described [84, 93]. The 96-well plates of mutant stocks were thawed completely, after which a 96-well transfer pin was used to transfer a small volume of mutants to a Nunc Omni Tray containing YPD agar with geneticin antibiotic. Between transfers, the 96-well transfer pin was sterilized with ethanol and flamed three times. After growing the cells for 48 h at 30˚C, the missing and slow growing mutants were recorded and two times the cell mass of these mutants were added separately. Working in a sterile hood, each tray was flooded with ~10 mL of YPD broth and all formed colonies were gently scraped by sterile cell spreader. The resuspended colonies were transferred into a sterile 1000 mL flask with a sterile stir bar. The suspension was mixed for 5 min on a stir plate to obtain an homogenized pool. The concentration of the freshly prepared pool was adjusted to 125–250 cells/mutant/μL by centrifugation at $500 \times g$. Once the concentration was adjusted, sterile glycerol was added to 15% (vol/vol) and 200 μL aliquots of the pool were stored in PCR strip tubes at −80˚C.

**Pooled competition with aurone SH1009.** Before exposing the pooled deletion mutants to SH1009, the inhibitory concentration for roughly 20% of the *S. cerevisiae* S288C wild type parent strain of these mutants was determined (500 μM) as previously described [93]. Using

96-well plates, 8 wells were filled with 198 μL of SH1009 diluted in YPD broth at concentration 500 μM with no more than 1% of DMSO. For the positive control, 8 wells were filled with 198 μL of MMS diluted in YPD broth at concentration 0.01 μg/mL. For the negative control, 8 wells were filled with 198 μL of YPD broth with 1% DMSO only. Two aliquots that were prepared from the previous step of the HOP deletion pool (~4,320 mutants), representing nonessential genes, and HIP deletion pool (~1,056 mutants), representing essential genes, were thawed completely. A 2 μL volume at a concertation of 125–250 cells/mutant/μL of the non-essential deletion pool were added to every 12 wells containing SH1009, MMS, and 1% DMSO. In the same manner, 2 μL of the essential deletion pool at a concertation of 125–250 cells/mutant/μL were added to every remaining 12 wells of SH1009, MMS, and 1% DMSO. After a 48h incubation at 30˚C, cells from each well were harvested independently by pipetting up and down and centrifuging at ~20,000 $x\,g$ for 3 min. The supernatant was removed and the pellet was processed for genomic DNA extraction.

**Construction of the DNA library.** Pellets of 24 samples were resuspended individually in 125 μL of Zymolyase solution and incubated for 1 h at 37˚C. The DNA was extracted from all 24 samples according to Maxwell 16 LEV Plant DNA Kit manual (Promega Corporation, Madison, WI, USA). To amplify the UPTAG unique 20 bp DNA barcodes as previously described [93], 24 PCR reactions were prepared independently, such that there was one PCR reaction for each sample in a total volume of 25 μL as follows: 21.5 μL of Taq mix, 0.5 μL of reverse common primer at 0.5 μM, 0.5 μL of indexed primer at 0.5 μM (for each sample, a distinct indexed primer was used, S5 Table), and 2.5 μL of genomic DNA at ~ 100 ng. PCR conditions were as follows: 5 min at 95˚C for an initial denaturation, followed by 30 cycles of 1 min at 95˚C, 30 s at 55 ˚C, 45 s at 68˚C, then, 10 min at 68˚C for a final extension. After PCR reactions, 25 μL of all PCR products were pooled together from individual PCR tubes into one tube library. This library was purified by separation on a 2% TAE agarose gel for 50 min at 120V. The desired band (267 bp) containing the amplified UPTAG DNA barcodes was cut and purified from the gel using a QIAGEN Quick Gel Extraction Kit (QIAGEN, Germantown, MD, USA). The library was diluted to 1:5,000, 1:10,000, and 1:20,000 and quantified with the KAPA Library Quantification Kit (KAPA Biosystems, Wilmington, MA, USA) and Bio-Rad CFX96 real-time PCR system (Hercules, CA, USA). After quantifying the correct concentration of the library, the library was prepared as a DNA template at the final concentration 15 nM with a 5% Spike-In of PhiX control according to the MiSeq System Denature and Dilute Library Guide (Illumina, San Diego, CA, USA). The Illumina MiSeq sequencer was used to run the DNA template for $1 \times 50$ cycles to yield a cluster density of 700–900 k/mm$^2$.

**Post-sequencing data analysis.** The Illumina sequencer generated a Fastq file that was converted to Fasta file using a converter tool due to the ease of manipulating the Fasta file [78]. In order to process and analyze sequence reads, Perl scripts were created (https://github.com/fma3b/Barcode_Seq_Analysis) [94]. Two raw-count files for HIP-HOP profiles were imported into excel sheet (Microsoft Corporation, US) to normalize the absolute counts and calculate fitness scores, Z-scores, *P*-values, and FDR values following calculations previously reported [84]. The raw sequence reads were deposited in Sequence Read Archive (SRA) under project number PRJNA491750. Enrichment analysis of KEGG pathway and gene ontology (GO) analysis was conducted using hypergeometric testing through ClueGo software to find the significantly enriched KEGG/GO terms using GO categories in the *Saccharomyces cerevisiae*-S288C as a background. To visualize the interactive annotation network between significant genes, ClueGO and CluePedia apps [29, 31] along with Cytoscape (Cytoscape Consortium, USA) were used.

## Flow cytometry

*C. albicans* SC5314 cells were grown at 30˚C until reaching exponential phase, diluted to ~1.2–$3\times10^6$ CFU/mL, and treated with SH1009 at the $IC_{50}$ concentration followed by incubation for an additional 3 h. After harvesting by centrifugation, supernatants were discarded and pellets were washed with PBS and then fixed with cold 70% ethanol at -20˚C for 2 h. Fixed cells were washed in PBS and resuspended in 500 uL of PBS containing 20 µg/mL RNase and incubated at 37˚C for 2 h. 200 µL of PBS containing 20 µg/mL propidium iodide (PI) were added to the treated cells. Using the Millipore Guava flow cytometer, 5000 events were counted, and the fluorescent intensity of PI measured. After acquiring the data using Guava PCA-96 software, the data was gated to exclude debris or aggregates. Experiments were performed in triplicate using cytochalasin D, which has been reported to arrest the cell cycle [35], as a positive control.

## Confocal microscopy

*C. albicans* SC5314 cells were grown at 35˚C until reaching exponential phase, then diluted to ~ 1.2–$3\times10^6$ CFU/mL. Aurone SH1009 was added at the $IC_{50}$ concentration and cells were incubated for 3 h. Cells were fixed by addition of 4% paraformaldehyde and subsequently incubated for 2 h at room temperature. After pelleting and washing the cells with PBS, cells were incubated with 1% Triton-X100 for 1 h at room temperature. Rhodamine Phalloidin (RP) was added to the cells followed by incubation in the dark at 4˚C for 1 h. After two PBS washes, cells were imaged using confocal microscopy (Zeiss, Thornwood, NJ, USA) at 60x magnification. Cells was assessed for actin distribution by considering that the actin is depolarized if more than five patches were observed in the mother cell [95]. Approximately 100 cells were counted per experiment in triplicate experiments. The *S. cerevisiae CDC42Δ* mutant was used as a positive control.

## RT-qPCR

After growing *C. albicans* SC5314 cells at 30˚C in YPD broth until reaching exponential phase, the culture was treated with aurone SH1009 at the $IC_{50}$ concentration followed by incubation for additional 1.5, 3, and 6 h. After harvesting the cells by centrifugation, the RNA was extracted according to the instructions of the Maxwell 16 LEV Plant RNA Kit. Total RNA from treated and untreated samples were normalized to 1 µg. cDNA was constructed by following the manufacturer protocol of SuperScript IV First-Strand Synthesis System kit (ThermoFisher, Waltham, MA, USA) using 10 ng of RNA. The RT-qPCR was preformed using 2× iQ SYBR green supermix (Bio-Rad, Hercules, CA, USA) under the recommended cycle conditions. All reactions were performed in triplicate using listed primer pairs (S6 Table). Transcript levels were normalized to the expression level of the housekeeping gene GAPDH and compared to the untreated sample using $\Delta\Delta C_T$ method [96].

## Supporting information

**S1 Table. Strains used in this study.**
(DOCX)

**S2 Table. Haploinsufficiency data analysis.**
(XLSX)

**S3 Table. Homozygous data analysis.**
(XLSX)

**S4 Table. Enrichment analysis of HIP-HOP profiles.**
(DOCX)

**S5 Table. Primers used in chemogenomic analysis.**
(DOCX)

**S6 Table. Primers used in RT-qPCR.**
(DOCX)

**S1 Fig. Growth rate of *S. cerevisiae* mutants.**
(TIF)

**S1 File. Synthesis of SH1009.**
(DOCX)

## Acknowledgments

We would like to extend our gratitude to Dr. Rebecca Seipelt-Thiemann for help with post-sequencing analysis and to Paola Alexandra Molina for assistance with confocal microscopy.

## Author Contributions

**Conceptualization:** Fatmah M. Alqahtani, Mary B. Farone.

**Data curation:** Fatmah M. Alqahtani, Mary B. Farone.

**Formal analysis:** Fatmah M. Alqahtani.

**Funding acquisition:** Scott T. Handy, Mary B. Farone.

**Investigation:** Fatmah M. Alqahtani, Brock A. Arivett.

**Methodology:** Fatmah M. Alqahtani, Brock A. Arivett, Mary B. Farone.

**Project administration:** Mary B. Farone.

**Resources:** Zachary E. Taylor, Scott T. Handy, Anthony L. Farone, Mary B. Farone.

**Software:** Fatmah M. Alqahtani.

**Supervision:** Scott T. Handy, Mary B. Farone.

**Validation:** Fatmah M. Alqahtani.

**Visualization:** Fatmah M. Alqahtani.

**Writing – original draft:** Fatmah M. Alqahtani.

**Writing – review & editing:** Fatmah M. Alqahtani, Anthony L. Farone, Mary B. Farone.

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
