## [Decision Letter · Decision Letter 0]

8 Aug 2019

PONE-D-19-18335

Chemogenomic profiling to understand the antifungal action of a bioactive aurone compound

PLOS ONE

Dear Dr. Farone,

Thank you for submitting your manuscript to PLOS ONE. After careful consideration, we feel that it has merit but does not fully meet PLOS ONE’s publication criteria as it currently stands. Therefore, we invite you to submit a revised version of the manuscript that addresses the points raised during the review process.

We would appreciate receiving your revised manuscript with in the next 3 months from this date. To enhance the reproducibility of your results, we recommend that if applicable you deposit your laboratory protocols in protocols.io, where a protocol can be assigned its own identifier (DOI) such that it can be cited independently in the future. For instructions see: http://journals.plos.org/plosone/s/submission-guidelines#loc-laboratory-protocols

We look forward to receiving your revised manuscript.

Kind regards,

Shankar Thangamani, DVM, PhD

Academic Editor

PLOS ONE

Journal Requirements:

3. In your Methods section, please give the sources of any cell lines and fungal strains used in your study.

4. We note that you have reported significance probabilities of 0 in places. Since p=0 is not strictly possible, please correct this to a more appropriate limit, eg 'p<0.0001'.

5. Our internal editors have looked over your manuscript and determined that it could be within the scope of our Antimicrobial Resistance call for papers. This collection of papers is headed by a team of Guest Editors for PLOS ONE: Kathryn Holt (Monash University and London School of Hygiene and Tropical Medicine), Alison H. Holmes (Imperial College London), Alessandro Cassini (WHO Infection Prevention and Control Global Unit), Jaap A. Wagenaar (Utrecht University). The Collection will encompass a diverse range of research articles; additional information can be found on our announcement page: https://collections.plos.org/s/antimicrobial-resistance. If you would like your manuscript to be considered for this collection, please let us know in your cover letter and we will ensure that your paper is treated as if you were responding to this call. If you would prefer to remove your manuscript from collection consideration, please specify this in the cover letter.

This research has been supported by funding from the Tennessee Center for Botanical Medicine Research at Middle Tennessee State University. We would also like to extend our gratitude to Dr. Rebecca Seipelt-Thiemann for help with post-sequencing analysis and to Paola Alexandra Molina for assistance with confocal microscopy.

The authors received no specific funding for this work.

Reviewers' comments:

Reviewer's Responses to Questions

**Comments to the Author**

1. Is the manuscript technically sound, and do the data support the conclusions?

Reviewer #1: Yes

Reviewer #2: Partly

2. Has the statistical analysis been performed appropriately and rigorously? 

Reviewer #1: Yes

Reviewer #2: Yes

3. Have the authors made all data underlying the findings in their manuscript fully available?

Reviewer #1: Yes

Reviewer #2: Yes

4. Is the manuscript presented in an intelligible fashion and written in standard English?

Reviewer #1: Yes

Reviewer #2: Yes

5. Review Comments to the Author

Reviewer #1: The authors used a standard microbiological techniques to assess the antifungal activity of aurone SH1009 and also implemented chemogenomic profiling in order to understand the mechanism of the antifungal activity of Aurone SH1009.

Major Comment: Antifungal drug discovery is immensely hampered by the scarcity of selective fungal targets due to the significant resemblance between mammalian and fungal cells. The chemogenomic profiling used in this study was not able to identify a molecular target that is specific to fungi, instead a plethora of metabolic and biological pathways were identified as the most affected. Unfortunately, these affected pathways are not specific to fungi and are shared with mammalian cells and hence their identification as a potential target for aurone doesn’t support the authors’ claims of potential novel target. In fact, the concluded mechanism of action is anticipated and typical for an anticancer compound. However, in order to consider this study as a significant contribution to the field of antifungal drug discovery, the authors should perform toxicity studies to demonstrate that aurone SH1009 is safe to mammalian cells at the tested concentrations. It would be important to see that aurone has a selective toxicity toward fungal cells and doesn’t produce similar effect on mammalian cell cycle, at least at the tested concentrations.

Minor Comments:

Abstract.

The epidemiological data provided in line 33 and 34 should be included in the introduction and be supported with references.

Introduction

Line 68: remove comma after candidiasis

Line 79: better replace the word invade with colonize

Line 87: the mode of action of amphotericin does not alter ergosterol, instead it bind to ergosterol and form pores in the cytoplasmic membrane … please modify the sentence

Paragraph 2: Why picking few resistance mechanisms adopted by fungi and ignoring other mechanisms without justification. For example there is at least four mechanisms of azole resistance and the authors just mention one mechanism (hyperexpression of efflux pumps). Same thing applies for amphotericin and echinocandin. I prefer to remove the mechanisms of resistance from the introduction part.

Result:

• Table 1: Please be consistent in your presentation of MIC data. Either include the standard error in both IC50 or MIC90 or remove it.

• Define MIC 90 in the title of table 1

It is better to stick to CLSI appreviation. Use MIC 50 instead of IC 50

Line 237, aurone at 400 uM represent 25 x MIC not 5 x MIC, based on the provided MIC data (16 uM)

ST1. Please replace the phrase (resistant to everything) and be specific, such as resistant to amphotericin B, fluconazole, itraconazole…..etc

M&M:

The author should follow the most recent vision of CLSI when performing the antifungal susceptibility testing. The referenced protocol M27A3 (2008) is old and the 2017 newer version is available.

Discussion

Line 736, I believe the author means ERG3 not ERG2. Please check.

Line 740, please remove the word (intrinsically) as the original study by Pfaller 2004 followed an older version of CLSI where the MIC data were recorded after 48h of incubation, hence most C. glabrata isolates appeared resistant to azoles. However recent reports indicate that Azole resistance in C. glabrata do not exceed 10 %. So, I think the word intrinsically is inaccurate description for the azole resistance in C. glabrata. Please review the article (https://doi.org/10.3389/fmicb.2016.02173) for more information.

Reviewer #2: Major comments:

1. Lines 215-216: “indicate that there is not regrowth of C. albicans 5314 in the presence of SH1009 above 6.25µM”. This statement is incorrect. The next higher concentration used in figure 3A is 12.5µm. There could have been regrowth in the concentration range between 6.25-12.5µM. The sentence should be corrected to read that there is no regrowth at ≥12.5µM.

2. Line 237 specifies that 5 times the IC50 for SH1009 is 400µM while line 976 (time-kill assay in methods) specifies the same is equivalent to 500µM. Please explain why these values are different. Also, lines 257-260 indicate that even fluconazole and amphotericin B were used at a concentration which is equivalent to 5 times the concentration that reduces the growth of C. albicans by 50%. In this case, please specify the IC50 for fluconazole and amphotericin B in the figure legend. Also indicate how 5 times the IC50 for SH1009 (against C. albicans) is 400µM when IC50 is only 16.28µM.

3. Line 415 is incorrect. Fig. ‘7’ and not ‘6’ depicts the growth curves of Sachharomyces mutants.

4. Line 413: What dilutions of SH1009 were used in testing the 12 mutants individually for differences in growth to S. cerevisiae? Please indicate.

5. Why is the growth of each mutant in Fig. 7 not tested in the presence of the same concentration (500 µM) as the one used in the HIP/HOP screens? Also, this reviewer thinks that this figure should include graphs showing the growth of mutants in the presence of 0 µM, 16 µM as well as 500 µM concentration of SH1009 to ensure comparable analysis.

6. Figure 9 should include panels with one sensitive (ARK1Δ) and one resistant (END3Δ) endocytic mutants identified through the HIP/HOP screening to show their effects on actin polarization.

7. Lines 604-605: Figure 7A shows the growth of mutants exposed to SH1009 at a concentration of 16 µM. Please explain how does the figure demonstrate that CDC42 deletion mutant is hypersensitive to aurone treatment at concentration of 3 µM?

Minor comments:

1. In Table 1, please write the names of Candida albicans strains M1-M7 in an increasing order (M4, M5 and M6) to enable easy analysis.

2. The replicates performed in the cell viability assay have not been specified in the figure 2. Please indicate the number of replicates and indicate the standard deviations for the different treatments in the dot plots itself. Also, please indicate the treatments above each dot plot in the figure so that that it is self explanatory.

3. Line 252: please rewrite the dilutions of SH1009 as “3.125 µM-200 µM”, the same as is written in the methods (line 964).

4. Label the figure 4A and 4B to indicate HIP or/ HIP profile so that the figure is self-explanatory.

6. PLOS authors have the option to publish the peer review history of their article (what does this mean?). If published, this will include your full peer review and any attached files.

Reviewer #1: Yes: Hassan Eldesouky

Reviewer #2: No

---

## [Author Response · Author response to Decision Letter 0]

3 Nov 2019

RESPONSE TO REVIEWERS: (Can also be found as a file attachment)

Note that the editor and reviewer comments appear as a numbered comment and in brackets.

Our responses appear below each comment.

Page and line numbers refer to the newly submitted “Manuscript” document.

Response to Academic Editor comments on Journal Requirements: 

Comment 1: [When submitting your revision, we need you to address these additional requirements. Please ensure that your manuscript meets PLOS ONE's style requirements, including those for file naming. The PLOS ONE style templates can be found at http://www.journals.plos.org/plosone/s/file?id=wjVg/PLOSOne_formatting_sample_main_body.pdf and http://www.journals.plos.org/plosone/s/file?id=ba62/PLOSOne_formatting_sample_title_authors_affiliations.pdf]

Response 1: We have, accordingly, changed the name of files to “Manuscript,” “Cover Letter,” and “Title Authors Affiliations.”

Comment 2: [We note that you have indicated that data from this study are available upon request. PLOS only allows data to be available upon request if there are legal or ethical restrictions on sharing data publicly. For more information on unacceptable data access restrictions, please see http://journals.plos.org/plosone/s/data-availability#loc-recommended-repositories. 

a) If there are ethical or legal restrictions on sharing a de-identified data set, please explain them in detail (e.g., data contain potentially sensitive information, data are owned by a third-party organization, etc.) and who has imposed them (e.g., an ethics committee). Please also provide contact information for a data access committee, ethics committee, or other institutional body to which data requests may be sent.] 

Response 2a: There are no ethical or legal restrictions on our sharing a de-identified data set.

[b) If there are no restrictions, please upload the minimal anonymized data set necessary to replicate your study findings as either Supporting Information files or to a stable, public repository and provide us with the relevant URLs, DOIs, or accession numbers. For a list of acceptable repositories, please see http://journals.plos.org/plosone/s/data-availability#loc-recommended-repositories.]

Response 2b: 

The sequence data are now publicly available in the SRA Run selector:

https://trace.ncbi.nlm.nih.gov/Traces/study1/?acc=PRJNA491750

Comment 3: [In your Methods section, please give the sources of any cell lines and fungal strains used in your study.]

Response 3: We have, accordingly, provided the sources of the mammalian cell lines and fungal strains. 

Materials and Methods, Materials and Reagents sub-section, Page 40, Lines 922 – 926. 

Comment 4: [We note that you have reported significance probabilities of 0 in places. Since p=0 is not strictly possible, please correct this to a more appropriate limit, eg 'p<0.0001'.]

Response 4: 

The correction to appropriate limits has been made in the following 6 positions: 

Results, Page 9, Line 204

Results, Page 14, Line 321

Results, Page 15, Lines 339, 343, and 347

Results, Page 16, Line 372

Results, Page 19, Line 424

Results, Page 22, Lines 507-508

Results, Page 26, Line 599

Comment 5: [Our internal editors have looked over your manuscript and determined that it could be within the scope of our Antimicrobial Resistance call for papers. This collection of papers is headed by a team of Guest Editors for PLOS ONE: Kathryn Holt (Monash University and London School of Hygiene and Tropical Medicine), Alison H. Holmes (Imperial College London), Alessandro Cassini (WHO Infection Prevention and Control Global Unit), Jaap A. Wagenaar (Utrecht University). The Collection will encompass a diverse range of research articles; additional information can be found on our announcement page: https://collections.plos.org/s/antimicrobial-resistance. If you would like your manuscript to be considered for this collection, please let us know in your cover letter and we will ensure that your paper is treated as if you were responding to this call. If you would prefer to remove your manuscript from collection consideration, please specify this in the cover letter.]

Response 5: We would be honored for our manuscript to be considered for publication in the collection of papers on Antimicrobial Resistance if it is still timely to do so.

Comment 6: [Thank you for stating the following in the Acknowledgments Section of your manuscript: This research has been supported by funding from the Tennessee Center for Botanical Medicine Research at Middle Tennessee State University. We would also like to extend our gratitude to Dr. Rebecca Seipelt-Thiemann for help with post-sequencing analysis and to Paola Alexandra Molina for assistance with confocal microscopy. We note that you have provided funding information that is not currently declared in your Funding Statement. However, funding information should not appear in the Acknowledgments section or other areas of your manuscript. We will only publish funding information present in the Funding Statement section of the online submission form. Please remove any funding-related text from the manuscript and let us know how you would like to update your Funding Statement. Currently, your Funding Statement reads as follows: The authors received no specific funding for this work.]

Response 6: 

We have removed the funding-related text from the Acknowledgments section and would like to alter our Financial Disclosure statement to reflect the funding.

Acknowledgements, Page 51, Lines 1160 - 1162. 

We would like the Financial Disclosure to be re-worded as follows:

This research was supported in part by the Tennessee Center for Botanical Medicine Research and the Molecular Biosciences Program at Middle Tennessee State University. The funders had not role in the study design, data collection and analysis, the decision to publish, or manuscript preparation.

Response to Reviewers' comments: 

Reviewer #1: 

Major Comment: [Antifungal drug discovery is immensely hampered by the scarcity of selective fungal targets due to the significant resemblance between mammalian and fungal cells. The chemogenomic profiling used in this study was not able to identify a molecular target that is specific to fungi, instead a plethora of metabolic and biological pathways were identified as the most affected. Unfortunately, these affected pathways are not specific to fungi and are shared with mammalian cells and hence their identification as a potential target for aurone doesn’t support the authors’ claims of potential novel target. In fact, the concluded mechanism of action is anticipated and typical for an anticancer compound. However, in order to consider this study as a significant contribution to the field of antifungal drug discovery, the authors should perform toxicity studies to demonstrate that aurone SH1009 is safe to mammalian cells at the tested concentrations. It would be important to see that aurone has a selective toxicity toward fungal cells and doesn’t produce similar effect on mammalian cell cycle, at least at the tested concentrations.]

Response: 

We agree with the reviewer and thank them for bringing up this aspect of our study. Therefore, cytotoxicity assays for aurone SH1009 were performed using three different types of human cell lines: HepG2, A549 and THP-1 cells. Accordingly:

The main results of cytotoxicity assay have been added to the abstract. 

Abstract, Page 2, Lines 41 - 42 

Note that in order to add this wording, a part of an introductory sentence was deleted to maintain the 300 word limit.

Cytotoxicity results have been added to the Results in the “Aurone SH1009 is selectively inhibitory for Candida spp.” sub-section. 

Results, Pages 11 – 12, Lines 241 - 262.

Table 2 has been added to the Results section, Page 12. 

Panel 3D, which is dose-response curve of human cell lines to SH1009, has been added to Figure 3 

Figure 3 legend, Pages 12-13, Lines 269 - 285.

Additional wording has been added to the Conclusion section.

Conclusion, Page 39, Lines 902-903 

Additional reagents have been added to the “Materials and reagents” section

Materials and Methods, Page 40, Lines 924 – 932 

Methodology for the cytotoxicity testing has been added to the Materials and Methods as subsection “Cytotoxicity Assay" - Materials and Methods, Pages 44 – 45, Lines 1015 – 1039

Reviewer #1 Minor Comments: 

Abstract: Comment: [The epidemiological data provided in line 33 and 34 should be included in the introduction and be supported with references.]

Response: 

As suggested by the reviewer, we have included epidemiological data in the introduction with its supported reference. 

Introduction, Page 5, Lines 101 -103. 

Introduction:

Comment 1: [Line 68: remove comma after candidiasis]

Response 1: The comma has been removed. Page 3, Line 68

Comment 2: [Line 79: better replace the word invade with colonize]

Response 2: As suggested by the reviewer, this word has been replaced. Page 4, Line 79

Comment 3: [Line 87: the mode of action of amphotericin does not alter ergosterol, instead it bind to ergosterol and form pores in the cytoplasmic membrane ... please modify the sentence.]

Response 3: 

As suggested by the reviewer this sentence has been modified. Page 4, Lines 87 – 88

Comment 4: [Paragraph 2: Why picking few resistance mechanisms adopted by fungi and ignoring other mechanisms without justification? For example there is at least four mechanisms of azole resistance and the authors just mention one mechanism (hyperexpression of efflux pumps). Same thing applies for amphotericin and echinocandin. I prefer to remove the mechanisms of resistance from the introduction part.]

Response 4: 

These mechanisms were chosen to be included because they pertain to the antifungal testing results. However, we agree with the reviewer that is unclear as to why only particular mechanisms of resistance were chosen. We have removed the specific resistance mechanisms from this section, but have retained the references for these and other mechanisms.

Page 4, Lines 87 – 98

Results:

Comment 1: [Table 1: Please be consistent in your presentation of MIC data. Either include the standard error in both IC50 or MIC90 or remove it.]

Response 1: The standard error values have been added to IC50 values. Page 8, Table 1 

Comment 2: [Define MIC 90 in the title of table 1]

Response 2: The MIC90 has been defined in the Table 1 legend. Page 8, Line 175 

Comment 3: [It is better to stick to CLSI abbreviation. Use MIC50 instead of IC50]

Response 3: 

We thank the reviewer for this suggestion, but we prefer to use IC50 because several multidisciplinary fields are more familiar with the IC50 term. 

Comment 4: [Line 237, aurone at 400 uM represent 25 x MIC not 5 x MIC, based on the provided MIC data (16 uM)]

Response 4: 

We apologize for this confusing error and thank the reviewer for their careful reading of the manuscript. We meant a five-fold increased concentration (~500 uM) relative to the IC50 concentration (16.25 uM) as calculated from the serial two-fold dilution to determine the IC50. This concentration was written correctly in the Materials and Methods section (Time-kill assay: Page 43, Lines 1006 - 1007), but we made the mistake in the Results section. 

We have tried to revise the above and related statements using language that clarifies the chosen concentration as follows: 

Results Pages 10 -11, Lines 229 - 231: “Treatment with SH1009 at a concentration representing an approximate five-fold increase of the IC50 calculated from the two-fold dilution series (500 uM)…”

Figure 3 legend, Pages 12 - 13, Lines 275 - 278: “…treated with concentrations five-fold higher than the IC50 concentrations of SH1009 (16.28 µM), fluconazole (0.5 µg/mL) and amphotericin B (0.25 μg/mL)”

Materials and Methods, Page 43, Lines 1006 - 1007: 

 “…was treated with an aurone concentration approximately five-fold higher (500 µM) than the dilution producing the IC50 of SH1009 (16.25 µM) in the two-fold dilution series.”

In Figure 3C, the concentration of aurone SH1009 has been changed from (400 uM) to (500 uM) as was correctly stated in the Materials and Methods section.

Comment 5: [ST1. Please replace the phrase (resistant to everything) and be specific, such as resistant to amphotericin B, fluconazole, itraconazole.....etc]

Response 5: 

We have revised the sentence and specified the resistance in Table 1 and in Supporting Materials, S1 Table.

Results, Page 8, Lines 178 – 179, Supporting Materials, S1 Table

Materials and Methods:

Comment 1: [The author should follow the most recent vision of CLSI when performing the antifungal susceptibility testing. The referenced protocol M27A3 (2008) is old and the 2017 newer version is available.]

Response 1: 

The 2017 newer version of CLSI protocol has been cited as follows: 

Results, Page 7, Line 151

Discussion, Page 30, Line 312

Materials and Methods, Page 41, Lines 954 and 959

Discussion:

Comment 1: [Line 736, I believe the author means ERG3 not ERG2. Please check.]

Response 1: 

We can understand why the reviewer would ask us to check again. However, the study mentioned a mutation in ERG2 gene as a reason for polyene resistance. 

Jensen RH, Astvad KMT, Silva LV, Sanglard D, Jørgensen R, Nielsen KF, et al. Stepwise emergence of azole, echinocandin and amphotericin B multidrug resistance in vivo in Candida albicans orchestrated by multiple genetic alterations. J. Antimicrob. Chemother. 2015;70(9):2551-5.

Comment 2: [Line 740, please remove the word (intrinsically) as the original study by Pfaller 2004 followed an older version of CLSI where the MIC data were recorded after 48h of incubation, hence most C. glabrata isolates appeared resistant to azoles. However recent reports indicate that Azole resistance in C. glabrata do not exceed 10 %. So, I think the word intrinsically is inaccurate description for the azole resistance in C. glabrata. Please review the article (https://doi.org/10.3389/fmicb.2016.02173) for more information.]

Response 2: 

We thank the reviewer for their insight on C. glabrata, and as the reviewer suggested, the word “intrinsically” has been removed. Page 33, Line 762.

Reviewer #2: 

Major Comments: 

Comment 1: [Lines 215-216: “indicate that there is not regrowth of C. albicans 5314 in the presence of SH1009 above 6.25μM”. This statement is incorrect. The next higher concentration used in figure 3A is 12.5μm. There could have been regrowth in the concentration range between 6.25-12.5μM. The sentence should be corrected to read that there is no regrowth at ≥12.5μM.]

Response 1: 

We agree with the reviewer and thank them for their careful review. The sentence has been corrected to read that there is no regrowth above 12.5 µM. Results, Pages 9 – 10, Lines 208 – 209

Comment 2: [Line 237 specifies that 5 times the IC50 for SH1009 is 400μM while line 976 (time-kill assay in methods) specifies the same is equivalent to 500μM. Please explain why these values are different. Also, lines 257-260 indicate that even fluconazole and amphotericin B were used at a concentration which is equivalent to 5 times the concentration that reduces the growth of C. albicans by 50%. In this case, please specify the IC50 for fluconazole and amphotericin B in the figure legend. Also indicate how 5 times the IC50 for SH1009 (against C. albicans) is 400μM when IC50 is only 16.28μM.]

Response 2a: 

Line 237 specifies that 5 times the IC50 for SH1009 is 400μM while line 976 (time-kill assay in methods) specifies the same is equivalent to 500μM. Please explain why these values are different. Also indicate how 5 times the IC50 for SH1009 (against C. albicans) is 400μM when IC50 is only 16.28μM.

We apologize for this confusing error and thank the reviewer for their careful reading of the manuscript. We meant a five-fold increased concentration (~500 uM) relative to the IC50 concentration (16.25 uM) as calculated from the serial two-fold dilution to determine the IC50. This concentration was written correctly in the Materials and Methods section (Time-kill assay: Page 43, Lines 1006 - 1007), but we made the mistake in the Results section. 

We have tried to revise the above and related statements using language that clarifies the chosen concentration as follows: 

 Results Pages 10 -11, Lines 229 - 231:

 “Treatment with SH1009 at a concentration representing an approximate five-fold increase of the IC50 calculated from the two-fold dilution series (500 uM)…”

Figure 3 legend, Pages 12 - 13, Lines 275 - 278: 

 “…treated with concentrations five-fold higher than the IC50 concentrations of SH1009 (16.28 µM), fluconazole (0.5 µg/mL) and amphotericin B (0.25 μg/mL)”

Materials and Methods, Page 43, Lines 1006 - 1007: 

 “…was treated with an aurone concentration approximately five-fold higher (500 µM) than the dilution producing the IC50 of SH1009 (16.25 µM) in the two-fold dilution series.”

In Figure 3C, the concentration of aurone SH1009 has been changed from (400 uM) to (500 uM) as was correctly stated in the Materials and Methods section (please see the edited Figure 3 included below).

Response 2b: 

"Also, lines 257-260 indicate that even fluconazole and amphotericin B were used at a concentration which is equivalent to 5 times the concentration that reduces the growth of C. albicans by 50%. In this case, please specify the IC50 for fluconazole and amphotericin B in the figure legend."

We have specified the IC50 concentrations of fluconazole and amphotericin B in the legend for Figure 3.

Results, Page 13, Line 278

Comment 3: [Line 415 is incorrect. Fig. ‘7’ and not ‘6’ depicts the growth curves of Saccharomyces mutants.]

Response 3: 

We have revised the number of figure to correctly be Figure 7. Results, Page 19, Line 438 

Comment 4: [Line 413: What dilutions of SH1009 were used in testing the 12 mutants individually for differences in growth to S. cerevisiae? Please indicate.]

Response 4: 

We have indicated the dilutions of SH1009 in the text.

Results, Page 19, Lines 436 – 437

Comment 5: [Why is the growth of each mutant in Fig. 7 not tested in the presence of the same concentration (500 μM) as the one used in the HIP/HOP screens? Also, this reviewer thinks that this figure should include graphs showing the growth of mutants in the presence of 0 μM, 16 μM as well as 500 μM concentration of SH1009 to ensure comparable analysis.]

Response 5a: 

"Why is the growth of each mutant in Fig. 7 not tested in the presence of the same concentration (500 μM) as the one used in the HIP/HOP screens?"

We appreciate the reviewer’s question. For individual mutant screening, the final concentration of cells of each S. cerevisiae mutant was only 2.5-0.5�103 cells/mL; hence, we used 2-fold serial dilutions (3.125 µM –200 µM) of SH1009. 

In the HIP/HOP screen, the cell concentration was much higher (1.25� 109 cells/mL) in order to accommodate all the mutant in the population (125-250 cells/mutant/µL). Therefore, the concentration of SH1009 that caused only 20% inhibition to this higher cell concentration was 500 µM. 

Response 5b: 

"Also, this reviewer thinks that this figure should include graphs showing the growth of mutants in the presence of 0 μM, 16 μM as well as 500 μM concentration of SH1009 to ensure comparable analysis."

We agree with the reviewer and did, in fact, test at additional concentrations. We have now included multi-panel graphs as supplemental Fig. S1, which show the growth of S. cerevisiae WT and mutants in the presence of 0 μM, 16 μM, and 200 μM of aurone SH1009. Figure S1 is also included below for your reference. 

Results, Page 19, Line 439

Supporting Information, Page 56, line 1435 (reference to Supporting Information)

Comment 6: [Figure 9 should include panels with one sensitive (ARK1Δ) and one resistant (END3Δ) endocytic mutants identified through the HIP/HOP screening to show their effects on actin polarization.]

Response 6: 

Thank you for this suggestion. It would have been interesting to further explore actin polarization in these S. cerevisiae mutants (ARK1Δ and END3Δ). However, the focus of this study was Candida albicans, and it seems slightly out of the scope of this paper. The main goal of this procedure was to show the effect of bioactive aurone SH1009 on actin polarization in C. albicans. However, the using of S. cerevisiae CDC42∆ mutant is because this mutant was hypersensitive to SH1009 and harbors a deletion for an actin-regulatory gene; hence, used as a control. The actin polarization of S. cerevisiae mutants (ARK1Δ and END3Δ) have been reported previously in the literature.

Ayscough KR. Coupling actin dynamics to the endocytic process in Saccharomyces cerevisiae. Protoplasma. 2005;226(1-2):81-8.

Gourlay CW, Ayscough KR. Identification of an upstream regulatory pathway controlling actin-mediated apoptosis in yeast. J Cell Sci. 2005;118(Pt 10):2119-32.

Comment 7: [Lines 604-605: Figure 7A shows the growth of mutants exposed to SH1009 at a concentration of 16 μM. Please explain how does the figure demonstrate that CDC42 deletion mutant is hypersensitive to aurone treatment at concentration of 3 μM?]

Response 7: 

Thank you for bringing this to our attention. Although we do have data for CDC42Δ deletion mutant growth at 3 μM, showing that the mutant was hypersensitive even at this concentration, the sentence has been revised to 16 μM to reflect the results as shown in Figure 7A. Results section, Page 27, Line 628 

Reivewer #2 Minor Comments: 

Comment 1: [In Table 1, please write the names of Candida albicans strains M1-M7 in an increasing order (M4, M5 and M6) to enable easy analysis.]

Response 1: 

We have written the names of Candida albicans strains in increasing order in Table 1. Results section, Page 8, Lines 174 – 179

Comment 2: [The replicates performed in the cell viability assay have not been specified in the figure 2. Please indicate the number of replicates and indicate the standard deviations for the different treatments in the dot plots itself. Also, please indicate the treatments above each dot plot in the figure so that that it is self-explanatory.]

Response 2: 

We have modified Figure 2 in order to include the number of replicates, the standard deviation, and the name of treatments above each dot plot. In the legend for Figure 2, we have included the number of replicates and standard deviations (n=3±SD) for panel 2D. Results section, Page 9, Lines 195 – 204 

Comment 3: [Line 252: please rewrite the dilutions of SH1009 as “3.125 μM-200 μM”, the same as is written in the methods (line 964).]

Response 3: 

We have revised the dilutions of SH1009 as the reviewer suggested. Results section, Page 12, Line 286. 

Comment 4: [Label the figure 4A and 4B to indicate HIP or/ HIP profile so that the figure is self-explanatory.]

Response 4: 

We have labeled Figures 4A and 4B as the reviewer suggested. 

The authors would like to thank the reviewers again for their thorough and insightful review of the manuscript. All of the comments were much appreciated.

---

## [Editor Report · Decision Letter 1]

20 Nov 2019

Chemogenomic profiling to understand the antifungal action of a bioactive aurone compound

PONE-D-19-18335R1

Dear Dr. Farone,

We are pleased to inform you that your manuscript has been judged scientifically suitable for publication and will be formally accepted for publication once it complies with all outstanding technical requirements.

With kind regards,

Shankar Thangamani, DVM, PhD

Academic Editor

PLOS ONE
---

## [Editor Report · Acceptance letter]

3 Dec 2019

PONE-D-19-18335R1 

Chemogenomic profiling to understand the antifungal action of a bioactive aurone compound 

Dear Dr. Farone:

I am pleased to inform you that your manuscript has been deemed suitable for publication in PLOS ONE. Congratulations! Your manuscript is now with our production department. 

With kind regards,

on behalf of

Dr. Shankar Thangamani 

Academic Editor

PLOS ONE